# Transient rheology of the Sumatran mantle wedge revealed by a decade of great earthquakes

Qiang Qiu [1,2], James D.P. Moore [1], Sylvain Barbot[1,2], Lujia Feng [1] & Emma M. Hill [1,2]

Understanding the rheological properties of the upper mantle is essential to develop a consistent model of mantle dynamics and plate tectonics. However, the spatial distribution and temporal evolution of these properties remain unclear. Here, we infer the rheological properties of the asthenosphere across multiple great megathrust earthquakes between 2004 and 2014 along the Sumatran subduction zone, taking advantage of decade-long continuous GPS and tide-gauge measurements. We observe transient mantle wedge flow following these earthquakes, and infer the temporal evolution of the effective viscosity. We show that the evolution of stress and strain rate following these earthquakes is better matched by a bi-viscous than by a power-law rheology model, and we estimate laterally heterogeneous transient and background viscosities on the order of ~$10^{17}$ and ~$10^{19}$ Pa s, respectively. Our results constitute a preliminary rheological model to explain stress evolution within earthquake cycles and the development of seismic hazard in the region.

[1] Earth Observatory of Singapore, Nanyang Technological University, Singapore 639798, Singapore. [2] Asian School of the Environment, Nanyang Technological University, Singapore 639798, Singapore. Correspondence and requests for materials should be addressed to Q.Q. (email: qiuqiang2012@gmail.com) or to S.B. (email: sbarbot@ntu.edu.sg)

Great earthquakes generate large stress perturbations across a wide area in both the adjoining crust and upper mantle. The rheological structure of the lithosphere and asthenosphere plays an essential role in governing the relaxation of stress[1], and thus may modulate the intensity and frequency of earthquakes[2,3]. The complex stress interactions between the lithosphere and asthenosphere, due to their rheological properties, control the postseismic strain and stress rates following great earthquakes[2,4]. These enhanced strain and stress rates have been observed to modify global seismicity over long time spans[3] and can also directly or indirectly trigger large earthquakes[2,5]. However, the spatial distribution and temporal evolution of these rheological properties are poorly understood, since the pressure, temperature and strain rates at these depths preclude in situ measurements[6–8]. The recording of many large earthquakes within the same geological setting, with multiple data sets encompassing high temporal resolution, affords us an opportunity to better constrain the rheology of the upper mantle.

Over the past decade, modern geodetic measurements have provided the opportunity to record how the surface of the crust moved, both during and after the earthquake. For example, the 2004 Mw 9.2 Sumatra–Andaman earthquake was the first giant earthquake that was observed with modern geodetic methods[9–11], providing detailed surface displacement measurements over the decade following the earthquake. Such a high resolution set of spatial and temporal geodetic measurements should have afforded tighter constraints on the rheological structure of the lower crust and upper mantle; however, numerous studies have proposed different rheological models to explain the data[12–19]. Whether the early postseismic deformation can be explained by afterslip only or combination of afterslip and viscoelastic relaxation; or whether the long-term deformation can be explained by combination of afterslip and background viscoelastic relaxation or afterslip with a transient viscoelastic relaxation is still debated. Thus, the rheological structure of the lower crust and upper mantle is still poorly understood. This is similar to the 2012 Mw8.6 Wharton basin strike slip earthquake where the postseismic deformation has been explained by both a linear Burgers rheology[20,21] and a nonlinear power-law rheology[4]. The reasons behind these various interpretations are in part due to the natural trade-offs between the multiple coexisting postseismic mechanisms, but largely because the significant trade-offs arising from choices made when forward modelling viscoelastic deformation. It is crucial to address these model uncertainties clearly and precisely, through comparison with joint inversions of afterslip and viscoelastic relaxation[22].

Here we probe the rheological properties of the asthenosphere across multiple great megathrust earthquakes between 2004 and 2014 along the Sumatran subduction zone (Fig. 1). They are four Mw ≥ 7.8 events including the 2004 Mw 9.2 Sumatra–Andaman[11], the 2005 Mw 8.6 Nias–Simeulue[23], the 2007 Mw 8.4 Bengkulu[24] and the 2010 Mw 7.8 Mentawai[25] earthquakes. We develop a novel approach (Methods section) to image the time evolution of afterslip on the megathrust and viscous strain in the mantle wedge constrained by both near-field continuous Sumatran Global Positioning System (GPS) Array (SuGAr), far-field continuous and campaign GPS measurement and far-field regional tide gauges (Figs. 1 and 2). We image the accelerated mantle wedge flow following these great earthquakes through direct inference of the temporal evolution of the effective viscosity, illuminating its transient behaviour. The rheology of the mantle arises from the complex interactions of pressure, temperature, stress, grain size, fluid and mineral content of the ambient rocks. We show that the transient flow following these earthquakes is well matched by a linear rheological model with bi-viscous behaviour at time scales from days to years, and we estimate the transient and background viscosities on the order of $\sim 10^{17}$ and $\sim 10^{19}$ Pa s, respectively.

## Results

### Kinematic models of afterslip and viscoelastic flow. Building self-consistent models of localised and distributed postseismic deformation is challenging partly due to the different physics involved in frictional sliding and viscoelastic flow. Traditionally, afterslip models are estimated following application of dynamic forward-modelled viscoelastic flow corrections to the data, but this approach relies on an assumed rheological behaviour of the bulk rocks. To relax these assumptions, we invert for the kinematics of afterslip and viscous strain (Supplementary Fig. 1) simultaneously. Our inversion approach also intrinsically captures the mechanical coupling between afterslip and viscoelastic flow, including adjustments due to afterslip induced by viscoelastic flow and vice versa. The combined near-field and far-field networks provide variable afterslip and viscous strain resolution along the megathrust (Supplementary Fig. 2), so we focus our discussion on the best-resolved features. Our estimates for the afterslip surrounding the coseismic rupture patches (Fig. 3a) are in general consistent with previous postseismic studies in this region[13,26–29]. Incorporating the coupling with viscoelastic flow generally results in more afterslip at shallow depths and less afterslip at greater depths on the megathrust[30] compared to afterslip-only inversions (Supplementary Fig. 3). Our model qualitatively explains both near-field and far-field measurements (Supplementary Figs. 4 and 5). As expected, the majority of the near-field postseismic displacements are driven by the frictional afterslip on the megathrust (Figs. 4 and 5), while the widespread viscoelastic flow in the mantle wedge contributes significantly to the vertical displacements not only for the near-field, but also dominates the far-field vertical postseismic displacements (e.g. the tide gauges along the coast of Malaysia and Singapore (Fig. 4a), and the GPS vertical components at some example stations (Fig. 4b); we also show the horizontal components of the early time stage in Supplementary Fig. 6).

Our approach can, in principle, disentangle the contributions of afterslip and viscoelastic flow, provided enough data are available. To assess the potential and limitations of this experimental setting, we compare our kinematic results with dynamic simulations at four stations (PSKI, LNNG, MKMK and LAIS) in the relatively well-resolved Bengkulu area. We conducted three stress-driven simulations assuming a gravitational, vertically stratified viscoelastic Earth using VISCO1D[31,32]. First we assumed the rheological structure adopted by ref. [16] for the 2004 Sumatra–Andaman earthquake (Burgers body, Simulation 1), second the same rheological parameters from ref. [16] but without the transient creep component (Maxwell body, Simulation 2), and finally a forward model with the inverted transient and steady-state viscosity from cuboid 2 (Burgers body, Simulation 3). The initial coseismic stress changes are calculated from ref. [24]. We show the predicted time evolution of the surface displacement, including contributions from both afterslip and viscoelastic flow following the 2007 Bengkulu earthquake (Fig. 5). We note that the vertical displacements provide a key discriminant between these two mechanisms e.g. station PSKI in Fig. 5c, often exhibiting the opposite sense of motion in the vertical. The surface displacements predicted by the dynamic model with our inverted rheological parameters (Simulation 3) and by the direct inversion are comparable with each other, with some minor differences, most likely due to the model employing a uniform rheology as opposed to the spatial variations discovered in the inversion. In addition, ref. [16] utilises a thinner elastic lid in the layered dynamic model than we have in our inversion; thus, it predicts larger surface displacement near stress concentration regions (Supplementary Fig. 7). Our modelled displacements sit half way between the Maxwell and the Burgers models, indicating some consistency with previous models of the region. The small differences in inferred viscosity may be due to our modelling assumptions. In particular, our inversion allows for the possibility

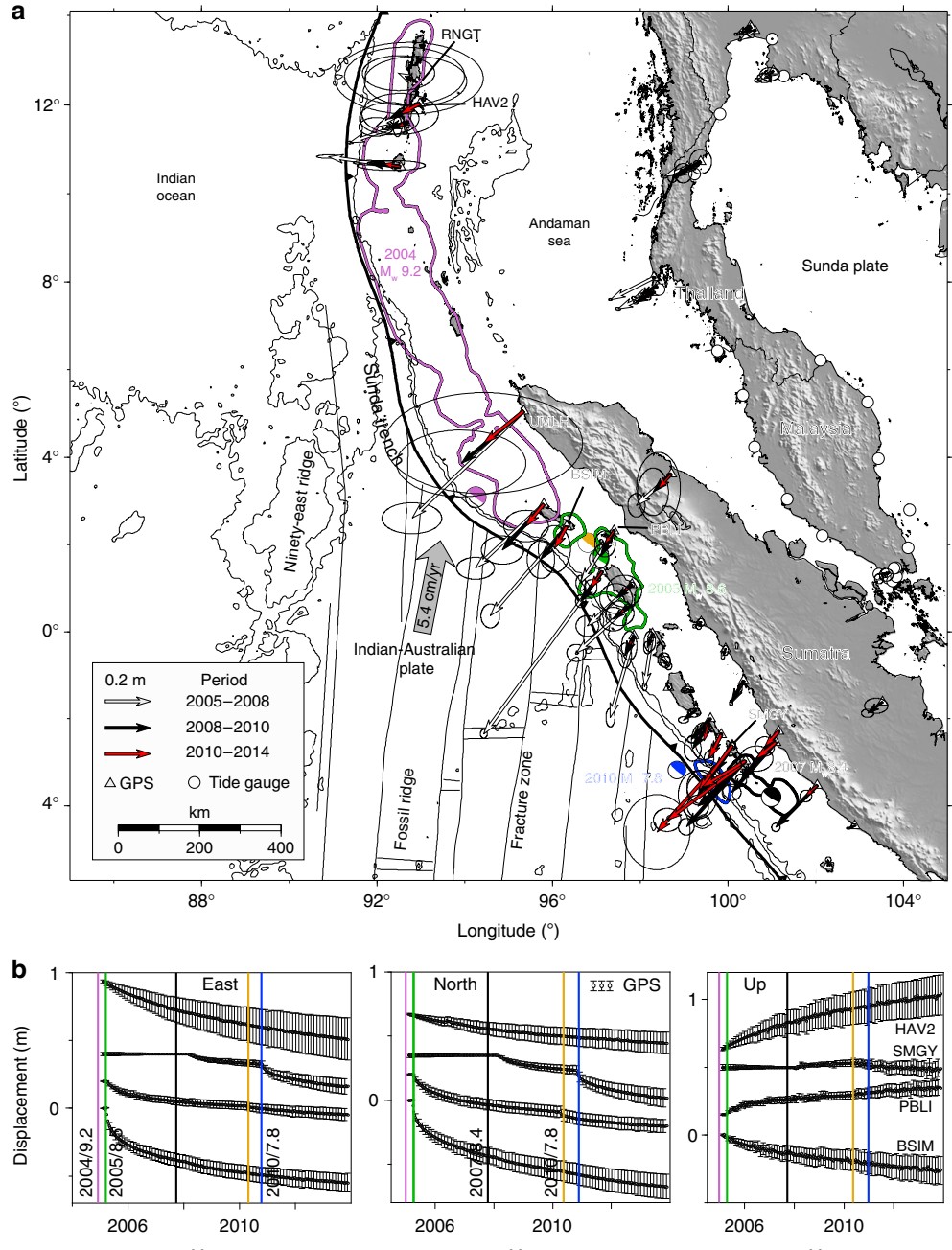

**Fig. 1** Cumulative postseismic displacements following great earthquakes of the Sumatran subduction zone from 2005 to 2014 as recorded by regional GPS stations. **a** Cumulative postseismic displacements over three separate time periods with 1σ error ellipses representing the 95% confidence intervals (note that for legibility we utilise the 60% confidence interval for sites UMLH and RNGT. Supplementary Fig. 4c illustrates the original 95% confidence intervals. All the GPS time series with the original "relative uncertainties" are shown in the Supplementary Fig. 4. Beach balls represent GCMT earthquake moment tensors of earthquakes, colour coded with event magnitude and times in **b**. Contours with matching colours to the moment tensors illustrate the 2 m slip contour for the estimated coseismic rupture patches for the 2004 $M_w$ 9.2 Sumatra-Andaman[11], the 2005 $M_w$ 8.6 Nias–Simeulue[23], the 2007 $M_w$ 8.4 Bengkulu[24] and the 2010 $M_w$ 7.8 Mentawai earthquake[25]. **b** GPS data (are plotted every ~40 days for clarity) at some example stations, illustrating the complex deformation associated with multiple great megathrust events over the last decade. The coloured vertical bars indicate the year and magnitude of great earthquakes

of arbitrary three dimensional (3D) heterogeneities, while previous approaches were restricted to vertical stratification and did not give afterslip a consistent treatment.

**Accelerated viscoelastic flow in the Sumatran mantle wedge.** We capture the acceleration of viscoelastic flow in the mantle wedge following the large megathrust earthquakes of the last decade until 2014 (Fig. 3a; Supplementary Movie 1). As suggested from the

resolution synthetic tests (Supplementary Fig. 2) and from the resolution matrix for the strain components $\varepsilon_{13}$ and $\varepsilon_{33}$ (Supplementary Figs 1 and 8), we are not able to estimate viscous strain and afterslip in the 2005 Nias–Simeulue rupture segment as accurately as the 2007 Bengkulu segment, due to the paucity of data directly down dip of the 2005 rupture above the viscous cuboids. Therefore, we focus on the best-resolved cuboids at the 2004 and 2007 rupture segments located in the shallowest portions of the megathrust (Fig. 3a,b). We track the temporal evolution of the stress and viscous

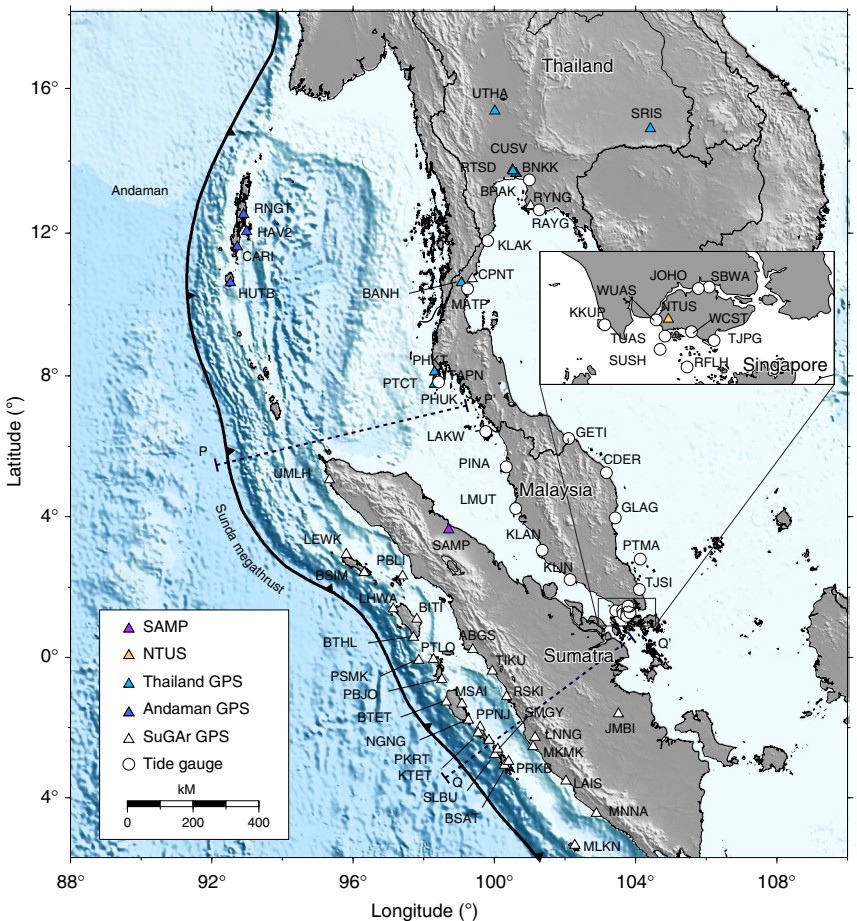

**Fig. 2** Geographic locations of all the near-field and far-field geodetic sites used in our study. White triangles indicate SuGAr stations. Purple triangle represents SAMP operated by the Indonesian National Coordination Agency for Surveys and Mapping. Blue triangles show GPS stations from ref. [12], where most of them are campaign sites established after the $M_w$ 9.2 2004 Sumatra–Andaman earthquake. Dark blue triangles are the four GPS stations on the Andaman Islands from ref. [28]. Open circles indicate tide gauges that provide the history of vertical land-height changes. The full name of the tide gauges are KKUP-Kukup, TUAS-Tuas, WUAS-West Tuas, SUSH-Sultan Shoal, WCST-West Coast, RFLH- Raffles Light House, TJPG-Tanjong Paga, SBWA-Sembawang, TJSI-Tanjung Sedili, PTMA-Pulau Tioman, KLIN-Tanjung Keeling, KLAN-Pelabuhan Kelang, LMUT-Lumut, GLAG-Tanjung Gelang, CDER-Cendering, PINA-Pulau Pinang, GETI-Getting, LAKW-Pulau Langkawi, TAPN-Ko Taphao Noi, MATP-Ko Mattaphon, KLAK-Ko Lak, JOHO-Johor Bahru, RAYG-Rayong and BPAK-Bang Pakong

strain rates (Supplementary Fig. 9a and b), and obtain the effective viscosity at each time step by dividing the estimated stress by the strain rate. The mechanical coupling between afterslip and viscoelastic flow is accounted for by keeping track of the time-dependent stress change induced by the mainshocks and the postseismic deformation (see Methods section, Supplementary Fig. 9a–c).

**Evolution of effective viscosity following earthquakes**. We directly evaluate the time series of effective viscosity from the kinematics of postseismic deformation. We see some consistency in the response of the mantle wedge following various megathrust earthquakes along the Sunda arc. We systematically infer a transient viscosity, beginning with smaller values (~$10^{17}$ to ~$10^{18}$ Pa s), and increasing by one order of magnitude within about 2 years (e.g. Fig. 6c,d). We note that temporal smoothing in the Kalman filter may bias the transient viscosity towards higher values, but smoothing is necessary to regularise the inversion. In subsequent years, the viscosities remain approximately constant (e.g. Fig. 6c,d). Such transient behaviour in the effective viscosity can be approximated to first order by a bi-viscous rheology. To extract the transient and steady-state viscosities, we fit our time series of effective viscosities with the exponential function

(Methods section)

$$\eta_{\mathrm{eff}}(t) = \frac{\eta_{\mathrm{K}}\eta_{\mathrm{M}}}{\eta_{\mathrm{M}}e^{\frac{-t}{\tau_{\mathrm{K}}}} + \eta_{\mathrm{K}}} \qquad (1)$$

The choice of the function is for convenience only, to identify the steady-state and transient viscosities, and does not imply any particular underlying physical mechanism. Considering one cuboid region at a time, we conduct a grid search for the best-fit values for the background viscosity $\eta_{\mathrm{M}}$, the transient viscosity $\eta_{\mathrm{K}}$, and the time scale of the transient $\tau_{\mathrm{K}} = \frac{\eta_{\mathrm{K}}}{G_{\mathrm{K}}}$, where $G_{\mathrm{K}}$ is the shear modular of the Kelvin body. Examples of the low-pass filtered time series of effective viscosities and best-fitting exponential model for cuboids 1 and 2 are shown in Fig. 6c,d (Supplementary Fig. 9c).

We note the reasonable agreement between the effective viscosity time series and the simplified model of Eq. (1) (e.g. Fig. 6c,d). Our estimated transient and background viscosities are on the order of ~$10^{17}$ Pa s and ~$10^{19}$ Pa s, respectively; with a characteristic time scale of $0.21 \pm 0.05$ yr for the transient. The time scale of the transient is commensurate with that found in other tectonic settings (e.g. ref.[33]) and our effective viscosity estimates from our example cuboids are also in agreement with

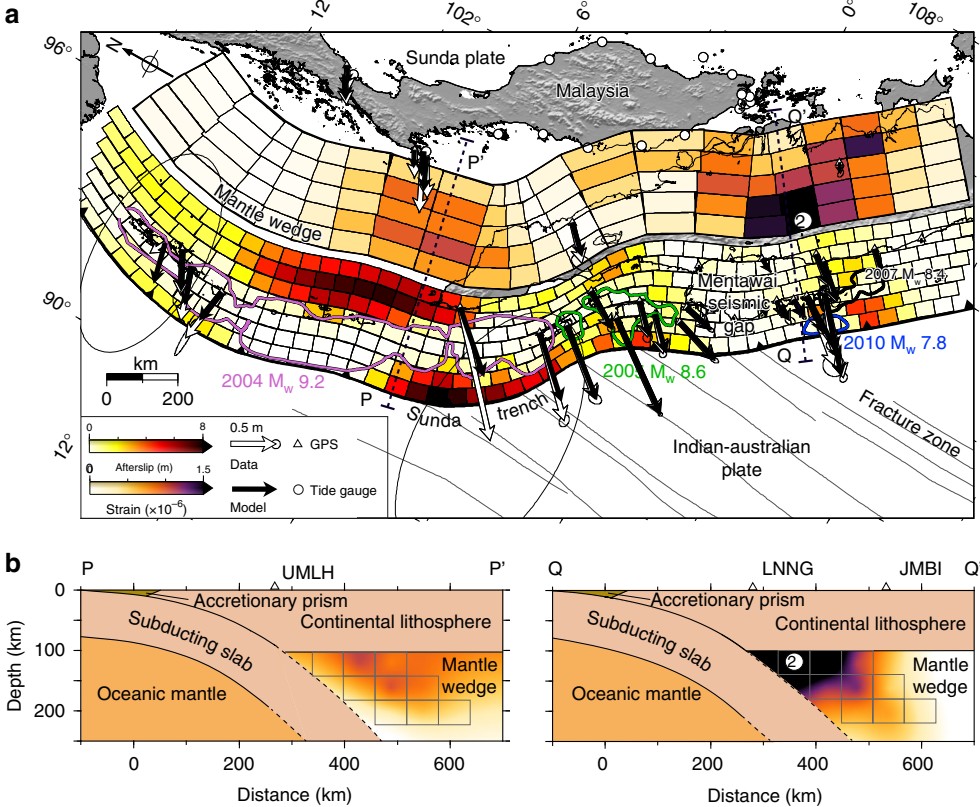

**Fig. 3** Cumulative afterslip surrounding the coseismic rupture areas and the second invariant of deviatoric viscous strain in the mantle wedge following these earthquakes from 2005 to 2014. **a** Cumulative total afterslip and the second invariant of deviatoric viscous strain for the shallowest cuboids in the mantle wedge. Coloured contours are the same as in Fig. 1a. Vectors show a comparison between predicted model displacements (black arrows) and observations (white arrows). Black dashed lines P–P′ and Q–Q′ show the locations of two cross-sections of the viscous strain estimated for the Sumatran–Andaman and Bengkulu rupture areas. **b** Cross-sections showing estimated viscous strain at P–P′ and Q–Q′ in **a**. The light grey boxes depict the ends of the cuboids

several published Burgers rheology models for the Sumatran subduction zone[12,14–16,18,34,35] (Supplementary Table 1 and Supplementary Fig. 10) despite the different modelling assumptions and techniques. These results imply that the time evolution of the viscosity is reasonably well recovered using our approach. The remaining differences can be attributed to the longer time series used in this study, and the fact that we do not constrain a priori the lateral and temporal variations in viscosity.

## Discussion

The full complexity of the rheology of the upper mantle originates from complicated interactions of temperature, pressure, stress levels, grain size, fluid and mineral content of the ambient rocks[4,8,19,36,37]. However, the gross behaviour can be investigated through a simplified theoretical framework involving transient and steady-state creep. In the 2004 Sumatra–Andaman rupture region, we image spatially decreasing transient (Fig. 6a and Fig. 7) and background viscosities towards Thailand (Fig. 6b). This low viscosity could be associated with backarc spreading around the Andaman Sea, as modelled by ref. [19], although we acknowledge the decreased spatial resolution in this region (Supplementary Fig. 2d). In the Mentawai segment, to the south, the steady-state viscosity is lower (Fig. 6b). We note the higher concentration of active volcanoes in the region, compared to farther north, which indicates a potential link between active volcanism and mantle mobility along the Sumatran arc[37].

To further address the constitutive behaviour of mantle rocks, we examine the in situ details of the stress–strain rate relationship at the location of our best-resolved cuboids (Fig. 8). The approach is similar to the analysis of laboratory data, where insight into the

rheological behaviour is gained through controlled experiments, so-called creep tests, where the strain rate is enforced by the apparatus and stress is measured during the deformation of a small sample. Our approach is similar, except for the boundary conditions and scale of investigation: the stress varies because of a large coseismic perturbation and the subsequent relaxation, and we interrogate an experimental volume of 240,000 km³.

The stress–strain rate phase diagram in log–log space (Fig. 8) reveals two distinct domains that we interpret as the signatures of transient creep and steady-state creep. Following both the 2004 Mw 9.2 Sumatra–Andaman earthquake (Fig. 8b) and the 2007 Mw 8.4 Bengkulu earthquake (Fig. 8a), the transient domain reveals an apparent nonlinear relationship that dominates the deformation for about 2 years, followed by a contrasting steady-state creep that obeys a near linear stress–strain rate relationship. These results are compatible with a combination of steady-state creep and transient creep, which may be attributed to background diffusion creep plus temporary motion of the soft slip system (grain boundary sliding during the transient phase) of the olivine crystals[4]. The stress/strain rate behaviour at cuboids 2 and 3 is well captured by the theoretical response of a Burgers material to a stress perturbation added to a background load (Fig. 8; Methods section), and we could not explain these data with a power-law rheology with a power exponent of 2 or greater. In addition, the estimated magnitude of the transient and steady-state viscosities of the resolved cuboids are not correlated with the magnitude of the spatial distribution of the coseismic stress changes (Supplementary Fig. 11a–c), further ruling out the possibility of the nonlinear power-law rheology. We conclude that a Burgers rheology involving linear work-hardening transient creep and

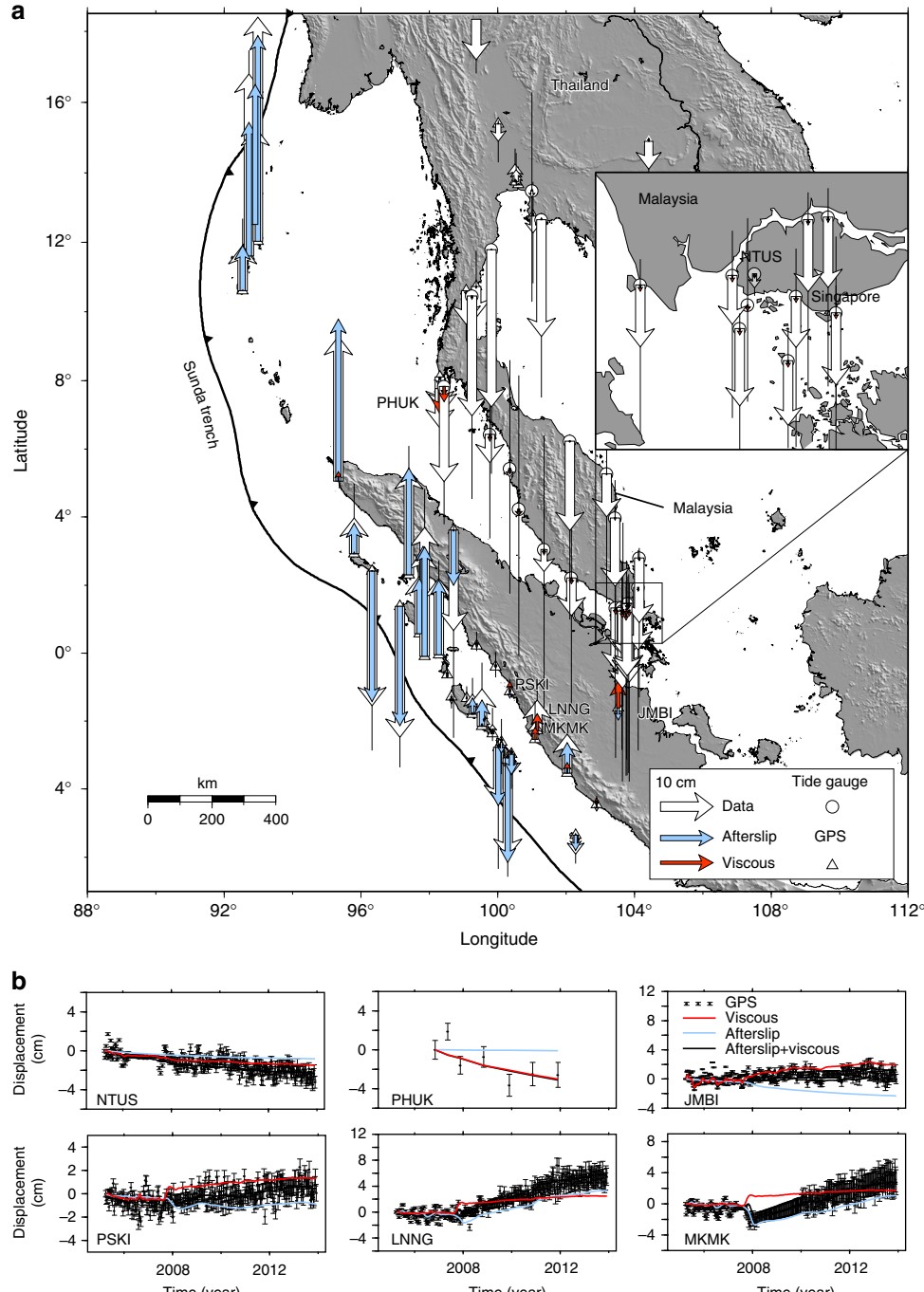

**Fig. 4** Our estimated cumulative displacements for the vertical component of the mechanical decomposition of afterslip and viscoelastic flow at all geodetic stations over the whole-time period from 2005 to 2014. **a** White vectors represent the cumulative vertical displacements at GPS stations, and tide gauges with 1σ error bars. Light blue vectors show the cumulative vertical displacements due to afterslip on the megathrust. Red vectors indicate the cumulative vertical displacements from viscoelastic flow in the mantle wedge. Sample stations (with names are labelled in the map) where the viscoelastic flow contributes significantly to the displacement are shown in more detailed time series. **b** Red, blue, black curves and black dots represent the viscoelastic flow, afterslip prediction, combination of afterslip and viscoelastic flow and vertical displacement with 1σ error bars, respectively. Due to the large variation of the tide gauge time series, we plot the cumulative displacements from least-squares fits, for the full time series of GPS and original tide gauges are shown in the Supplementary Figures 4 and 5

linear steady-state creep is a realistic description of mantle rock rheology within a time scale of days to years. We anticipate that future geodetic data sets, with greater spatial and temporal resolution both inland and offshore, will help us to gain more insight into the rheological structure of the region.

The transient rheology of the upper mantle has been widely proposed to explain postseismic deformation following large earthquakes at other subduction zones[30,38], and also in intra-continental tectonic settings[39,40]. Our study highlights that at least two creep mechanisms accommodate the viscoelastic flow of the asthenospheric upper mantle, which can be captured by a linear bi-viscous Burgers rheology, at least to first order. Our findings provide new insight into the rheology of the mantle wedge, contributing to assessing seismic hazards and their

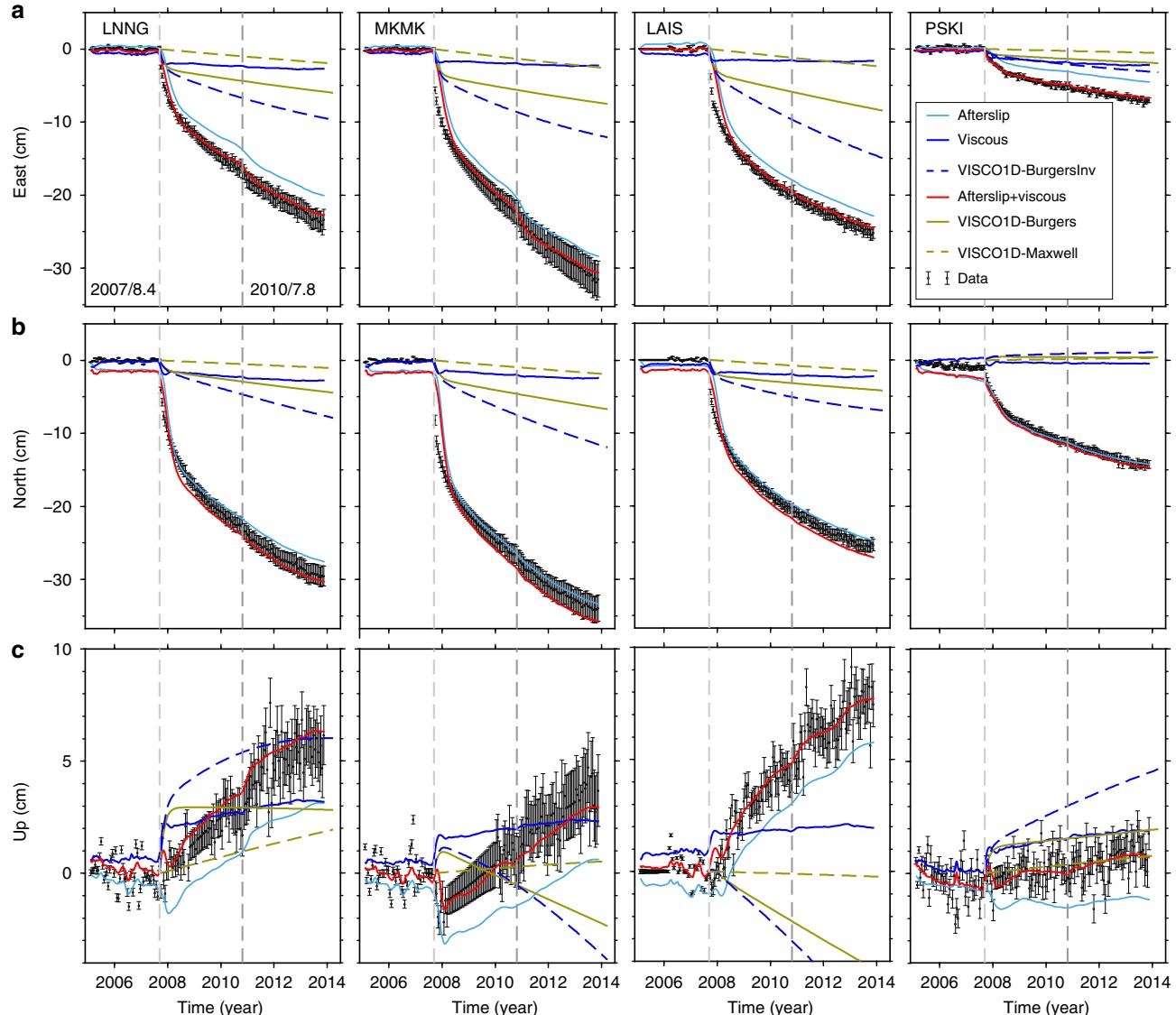

**Fig. 5** A comparison of GPS time series and model predictions. Predictions are from both afterslip and viscoelastic flow from our inversion and dynamic VISCO1D forward model at sample stations (see Figs. 2 and 4). **a**, **b**, **c** The east-component, north-component and up-component displacements, respectively. Grey dots show the GPS data with 1σ error bars. Afterslip and viscoelastic flow predictions from our inversion are shown in light blue (solid curves) and blue (solid curves), respectively, with their combined contribution shown in red. The forward model prediction by VISCO1D driven by the coseismic model from ref. [24] with the both Burgers, and Maxwell rheology model found at the 2004 rupture from ref. [16], and transient and steady-state viscosity found at cuboid 2 (BurgersInv) are respectively shown as brown curves, brown dashed lines and blue dashed lines for reference

associated tsunami hazards, over earthquake cycles in the Sumatran region.

## Methods

**GPS time series**. The postseismic deformation following each of these events is well captured by the near-field continuous Sumatran Global Positioning System (GPS) Array (SuGAr), far-field continuous and campaign GPS measurement, and far-field regional tide gauges (Figs. 1 and 2), resulting in a decade-long temporally dense time series. The combination of both near-field and far-field data sets provides us favourable conditions to infer the probable rheology of the mantle wedge in this region.

We use continuous time series from 29 stations of the SuGAr network, 1 GPS station (SAMP) from the Indonesian National Coordination Agency for Surveys and Mapping, 1 International Global Navigation Satellite System (GNSS) Service (IGS) site (NTUS) in Singapore, 4 stations on the Andaman Islands[28] and 12 sites in Thailand[12]. We also derive the far-field vertical land elevation changes at 23 tide-gauge locations around Singapore, Malaysia and Thailand, through a combination of tide-gauge data and sea-surface height altimetry Aviso time series. The resulting time series of displacements span the time period from early 2005 through to 2014,

providing constraints for the temporal evolution of the rheological parameters of the various postseismic relaxation mechanisms.

We process the GPS data up to the end of 2013 from 29 SuGAr stations, 1 IGS station (NTUS), 1 station (SAMP) operated by the Indonesian National Coordination Agency for Surveys and Mapping and 4 stations on the Andaman Islands[28] using the GPS-Inferred Positioning System and Orbit Analysis Simulation Software (GIPSY-OASIS) version 6.2[41]. In addition, we use the postseismic displacement at 12 GPS sites in Thailand from ref. [12]. Further details about our GPS processing methodology can be found in ref. [42]. We transform daily positions in the International Terrestrial Reference Frame 2008[43] (ITRF2008) to the Sunda plate reference frame using the ITRF2008-Sunda transformation[44]. Locations of the GPS stations are shown in Figs. 1 and 2.

For our modelling, we use GPS time series that contain only the postseismic signals from the recent multiple Sumatran earthquakes. To obtain the postseismic time series, we remove other signals, i.e. long-term rates, annual and semi-annual seasonal signals, coseismic offsets from all the megathrust events, and also the coseismic and postseismic displacement of the Mw 8.6 2012 Indian Ocean earthquake, which was not a megathrust event. All these signals are simultaneously estimated using a single-weighted nonlinear least-squares optimisation procedure for each station. The resulting estimates are used to not only remove non-postseismic

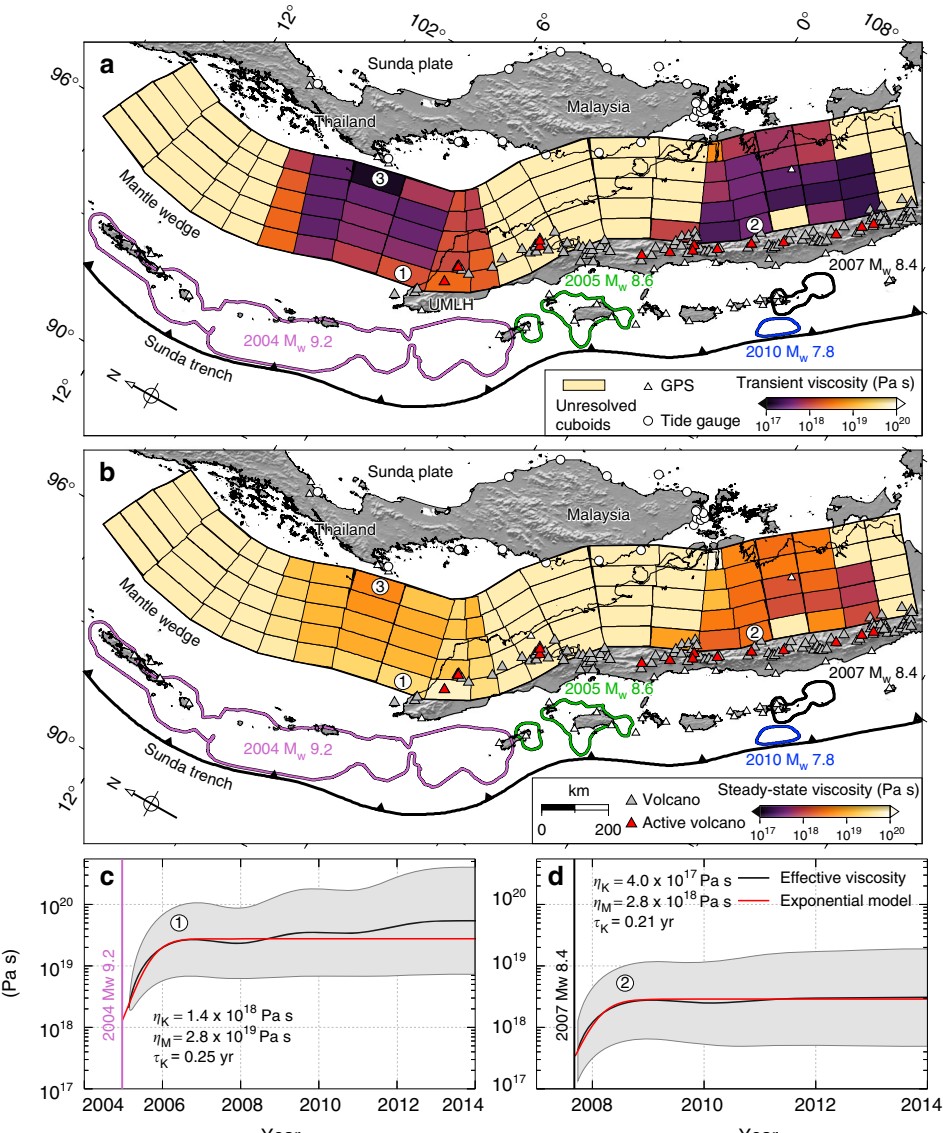

**Fig. 6** Effective viscosities derived from the bi-viscous Burgers rheological model with transient Kelvin and background or steady-state Maxwell viscosity estimates. **a** Colours indicate the inferred Kelvin viscosity of each cuboid. Cuboids lacking sufficient resolution to provide a robust viscosity estimate are filled with pale yellow and labelled as unresolved cuboids. Slip contours of the great earthquakes are the same as Figs. 1 and 2. We select two of the better-resolved cuboids (labelled (1) and (2), where (2) and (3) are also illustrated in Fig. 8) to illustrate the temporal evolution of effective viscosity in **c**, **d**. Volcanoes on Sumatra are represented by grey triangles and active ones shown as red. **b** Same as **a** but for the inferred Maxwell viscosities. **c,d** Time evolution of effective viscosity at cuboids 1 and 2 in **a** and the best-fit equivalent Burgers rheological model, respectively

signals but also to fill in data gaps. Further details about the signal estimation procedures are described in ref.[42].

**Land-height displacement derivation**. Tide gauges measure the relative sea level between the sea surface and the land. Thus, tide gauges capture changes of the land elevations. To extract the land-height changes, information on sea-surface changes are required. Satellite altimetry measurements can provide this information. A combination of tide-gauge and satellite altimetry data has been widely used to extract land-height changes[45–48]. Land-height changes following the 2004 Sumatra–Andaman earthquake were successfully extracted from tide gauges by ref.[49].

We carefully select tide-gauge data that have as few gaps as possible because multiple gaps or shorter tide recordings can bias the long-term trend estimation. Our final selected tide gauges are located in Singapore, Malaysia and Thailand, covering a wide area in the far-field of the Sunda megathrust (Figs. 1 and 2). We downloaded the tide-gauge data from the Permanent Service for Mean Sea Level database[50] (PSMSL, 2016, from website http://www.psmsl.org/data/).

For the satellite altimetry data, we use the Aviso data product from ftp.aviso.altimetry.fr. The altimeter products were produced by Ssalto/DUACS and distributed by Aviso, with support from Cnes (http://www.aviso.altimetry.fr/duacs/). The Aviso data are from a combination of all altimeter missions, and have been corrected for

instrument, orbit errors and errors resulting from signal propagating through the atmosphere, as well as perturbation errors from surface reflection. Tides, such as pole tides, solid, ocean and loading tides are corrected as well. Cross-calibration is also applied to yield a consistent and homogenous high-quality data set[51]. This data set has been successfully and widely implemented for sea level studies[45,52–57]. We use the delay-time mode of the global mean sea level anomalies product of the Aviso data set because of its accuracy as compared with real-time mode. These global, gridded, multi-mission data span from 1993 until May 2014 with daily solutions. In order to extract the land-height information by subtracting altimetry data with the monthly tide-gauge data, we use the daily solutions to estimate monthly average sea surface height anomalies.

A previous study in this region from ref.[49] reports that if the distance between the tide gauge location and the satellite observation grid data is less than 100 km, a good estimate of land-height changes can be achieved. Using this criterion, we select 24 tide gauges with nearby Aviso grid points to extract the land-height changes following the megathrust earthquakes at the Sumatran subduction zone between 2004 and 2014.

Before subtracting the Aviso data from the tide-gauge data to obtain the land-height changes, we remove seasonal cycles from both data sets. To do this, we use a simple but robust technique as described by ref.[58]. We average the whole-time period data for each month, and then subtract the monthly averages from each month of the time series. This technique is better than a least-squares approach,

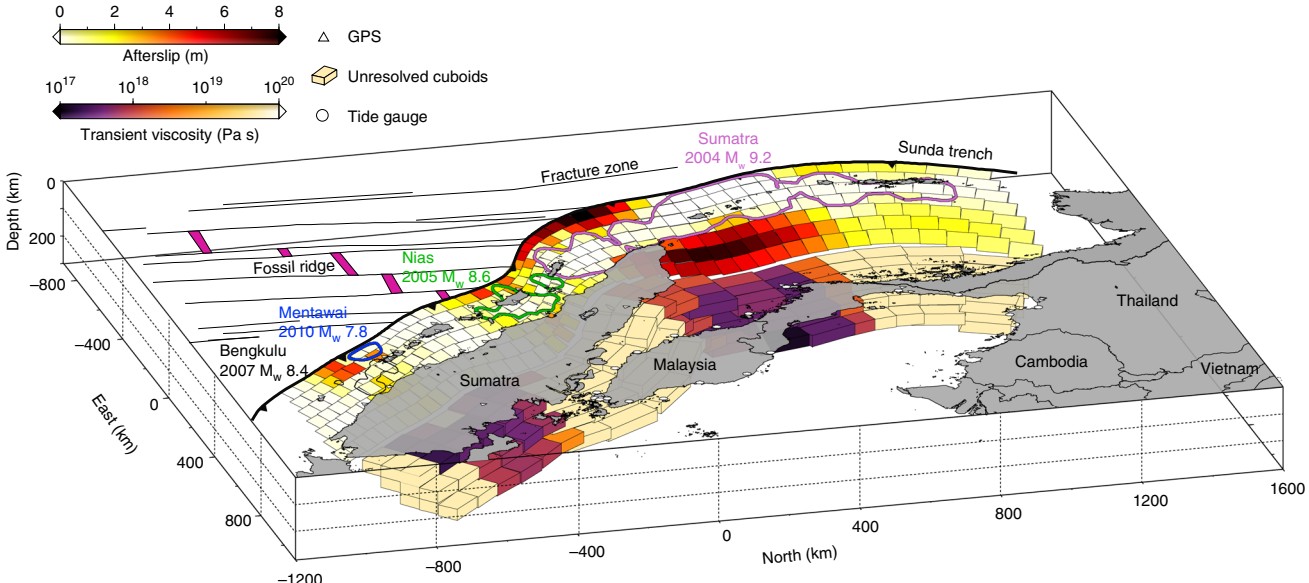

**Fig. 7** A 3D view of the cumulative afterslip surrounding the coseismic rupture areas from 2005 to 2014. Transient effective viscosities immediately following megathrust earthquakes along the Sumatran subduction zone. Coloured contours are the same as Figs. 1a, 3a and 6a,b, with the name and magnitude of earthquakes labelled the same colour code. Colour bars indicate the afterslip and transient effective viscosity, respectively. Coloured patches and cuboids are the 3D presentation of megathrust and discretised mantle wedge at this region. Unresolved cuboids are labelled as light orange colour. Subducting seafloor structures, fracture zone and fossil ridge are represented as light black lines and magenta shaded areas, respectively

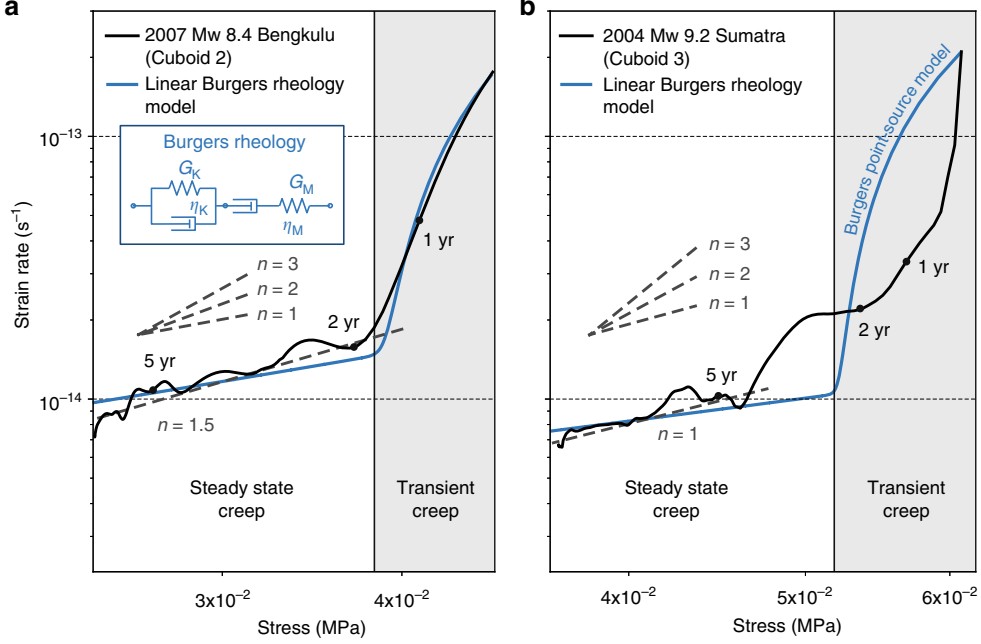

**Fig. 8** Strain rate and stress relationship for well-resolved example cuboids 2 and 3. **a,b** Cuboid 2 locates directly down dip of the 2007 $M_w$ 8.4 Bengkulu earthquake, and cuboid 3 sits down dip of the 2004 $M_w$ 9.2 Sumatra–Andaman earthquake, respectively (Fig. 6). Grey dashed lines represent the power exponents of strain rate to stress. The black dots indicate the state 1, 2 and 5 years after the earthquakes. The blue curves are the theoretical response of a Burgers spring-dashpot model under constant loading and a stress perturbation

because the least-squares method estimates constant seasonal parameters for the whole-time period of each tide gauge. Using this technique we find that the seasonal signals for each month can vary considerably. We show one example of seasonal signals from both Aviso and the tide gauges at Johor Baru (location shown in Fig. 2) in Supplementary Fig. 12.

We minimised the uncertainty by carefully removing the variable seasonal signals, and by finding the collocated tide gauge and Aviso grids pairs. However, there are a number of uncertainties that are still contained within the resultant land-height time series due to the low-spatial resolution of the Aviso grids. For example, the local complexities of shallow bathymetric effects and manmade land-

height signals that would be captured by the tide gauges are absent in the coarse altimetry Aviso grids. These localised signals would then be mapped into the land-height time series, potentially producing abnormalities in the land-height variation when compared to neighbouring tide gauges that were less affected by the local signals. For example, the high rate of subsidence at tide gauge BPAK-Bang Pakong (Fig. 2) is comparable or even larger than that of tide gauge TAPN- Ko Taphao Noi (Fig. 2), which was closer to the 2004 Sumatra–Andaman earthquake, and strongly affected by the postseismic deformation process (Supplementary Fig. 5). This abnormally high rate of subsidence likely contains contamination from human-induced subsidence at Bangkok, Thailand (see refs. [59,60]), ~50 km away. Tide-gauge

RAYG-Rayong (see Fig. 2) shows strong variations preceding the 2004 Mw 9.2 Sumatra–Andaman earthquake (Supplementary Fig. 5). Similar abnormal signals were also found at tide gauges east of Malaysia, which were less affected by the 2004 rupture. We also noted a systematic scatter in the vertical displacement time series for tide gauges around Singapore over time period ~2010 to 2011 (Supplementary Fig. 5). We are not clear what may have caused this offset, but we saw significant amplitude decline of the Aviso time series over this time period, which results this offset across all the tide gauges in Singapore region. To completely and accurately deal with these abnormal signals and uncertainties presents a substantial challenge. However, the tide gauge data do clearly show changes in their linear trend after the great megathrust events (vertical dashed lines in Supplementary Fig. 5), which provides a strong constraint on the direction of land movement, if not its magnitude, affording a crucial constraint for the mechanisms of the postseismic relaxation processes (see red profiles in Supplementary Fig. 5).

**Geometry of the fault and mantle wedge**. Our fault geometry is derived from a model used by ref. [15], who modified and extended the Slab 1.0 model to 16° North along the Sunda megathrust. We place the strain volumes (cuboids with ~100 km length, 60 km width and 40 km height) in the locality of the mantle wedge with a depth range from ~50 to ~230 km, consistent with previous postseismic studies of this region[12–16,18,34,35,61], with strike aligned to the strike of the megathrust. We ignore the viscoelastic relaxation of the oceanic asthenosphere, which will have limited influence on our observations over the time frame considered in this work[4].

**Joint inversion with the Extended Network Inversion Filter**. With both the near-field and far-field postseismic time series (observed motions corrected for interseismic, seasonal and other earthquake-related deformations), we invert for localised slip on the megathrust and distributed strain in finite volumes[22,27,62] (Supplementary Fig. 1), corresponding to afterslip on the megathrust and ductile flow in the mantle wedge, respectively. We do not consider localised and short-term deformation generated by pore pressure changes (poroelastic rebound[63]) due to coseismic stress changes, since we are focusing on long-term and widespread postseismic relaxation. We apply a modified version of the Extended Network Inversion Filter[64,65] (ENIF) to the time series data to infer the temporal evolution of afterslip on the megathrust and viscous strain in the mantle wedge.

The ENIF simultaneously inverts for both fault slip and viscous strain within a finite cuboid volume at asthenosphere depths. The original version only inverts for fault slip, whereas the modified version can invert for both fault slip and strain within a finite volume. Our modified ENIF models the cumulative surface displacement ($\mathbf{u}_i(r_j, t)$) of GPS sites at location $\mathbf{r}_j$, time $t$, as

$$\mathbf{u}_i(r_j, t) = \sum_k \int \mathbf{s}_k(\xi, t) G_{ik}(r_j, \xi) \mathrm{d}A(\xi)$$

$$+ \sum_{ml} \int\int\int \boldsymbol{\varepsilon}_{ml}(\vartheta, t) G_{iml}(R_j, \vartheta) \mathrm{d}V(\vartheta)$$

$$+ \mathbf{L}(\mathbf{r}_j, t) + \mathbf{f}(t) + \mathrm{E}$$

$$ml \in \{11, 12, 13, 22, 23, 33\}, \quad k \in \{1, 2\}, \tag{2}$$

where Einstein summation notation is implied and ($\mathbf{u}_i(r_j, t)$) represents the $i$th component of displacement at time $t$. The first term on the right hand side is the surface displacement at time $t$ due to slip $\mathbf{s}_k(\xi, t)$ on a fault patch $A(\xi)$, where $G_{ik}(\mathbf{r}_j, \xi)$ represents the Green's functions that map unit slip on a fault patch to surface displacement at a location $\mathbf{r}_j$. The second term is the surface displacement at time $t$ due to strain component $\boldsymbol{\varepsilon}_{ml}(\vartheta, t)$ of a cuboid $V(\vartheta)$ located at $\mathbf{R}_j$ in the subsurface. Each cuboid has six independent components of strain (i.e. $ml = 11$, 12, 13, 22, 23, 33) (Supplementary Fig. 1). $G_{iml}(R_j, \vartheta)$ are the 3D Green's functions that map the unit strain components to the surface displacement. $\mathbf{L}(r_j, t)$ are the local non-tectonic random benchmark motions e.g. the local motion of GPS monument. $\mathbf{f}(t)$ is the common mode error of the whole-GPS network such as seasonal loading, ocean loading, hydraulic loading, etc. E contains the observation errors that are not taken into account during the GPS processing (e.g. multiple path errors and azimuthally varying path delays). This error is assumed to follow a Gaussian distribution with zero mean and covariance $\sigma^2\Sigma_n$, where $\Sigma_n$ is the GPS data covariance matrix derived from GPS data processing.

We discretise the fault plane into rectangular sub-patches. We calculate the Green's functions (surface displacement at GPS stations) due to unit slip on the sub-patches, assuming that they are buried in a homogeneous, isotropic, elastic half-space, using the analytic solution of Okada[66]. We calculate the 3D Green's functions that map the strain components of the deformable cuboids to surface displacement at GPS stations through a novel analytic solution[62]. For each cuboid, there are six independent components of strain. We show an example of surface displacement associated with a single cuboid in Supplementary Fig. 1. To reduce computational burden and remain within the limit of the computational capacity, we down-sample the daily solution time series of both GPS and tide gauge data to every ten days. The instantaneous rates (e.g. slip and strain) might be affected

slightly; however, the final accumulated slip on the megathrust and strain in the mantle wedge should remain unchanged.

**Afterslip penalisation and spatial and temporal smoothing**. Imaging the spatial distribution of afterslip with two directions of slip on fault patches, and ductile flow with six components of strain within cuboid volumes (Supplementary Fig. 1) results in a glut of free parameters in the inversion. To avoid over-interpreting the available data, we apply extra constraints for both the afterslip and viscous strain to regularise them moving in a sensible way.

Afterslip is the continuation of slip as accelerated creep following the coseismic motion, upon which we enforce a no-backslip or positivity constraint (see ref. [65]). In addition, we apply spatial smoothing to both afterslip and strain components with a Laplacian matrix to minimise large jumps between neighbouring patches. The spatial and temporal smoothing hyper-parameters are estimated together with the model parameters for each model run, which is also one of the advantages of the ENIF[64,65] technique. The Kalman filter is a Bayesian filter, where the parameters are predicted through a transition matrix and updated with the measurement from the current step. Then we step through to the next time epoch. As more and more information is incorporated into the system through time, the estimates of the parameters consistently improve. The filter initialises with a prescribed starting value, then adjusts it within the specified uncertainty to improve the fit to the measurement. There is a trade-off between the spatial and temporal smoothing hyper-parameters. To address this potential issue, we conducted a few preliminary runs to determine a reasonable spatial smoothing hyper-parameter that reliably produces both a smooth slip distribution and good fit to the data. We then employ this value with a small covariance, thereby only allowing its value to be minimally changed by the Kalman filter, while using a large covariance for the temporal smoothing hyper-parameter which allows for a broad range of adjustments when necessary during the updating step. We finally obtain the initial guess for the temporal smoothing hyper-parameter, through a recursive algorithm where each previous estimate is updated to the current input, until model results from the current estimates and previous ones converge to within a specified tolerance (see ref. [65]).

**Isotropic strain and strain directions penalisation**. Viscoelastic flow in the asthenosphere is assumed to be deviatoric. To model the viscous strain, we penalise isotropic strain as well as strain directions that are perpendicular to the induced coseismic stress changes at each strain volume. Accordingly, we penalise the trace of the viscous strain tensor of each cuboid, encouraging, but not strictly enforcing it to be zero:

$$\mathcal{E}_{11} + \mathcal{E}_{22} + \mathcal{E}_{33} \approx 0. \tag{3}$$

Here $\mathcal{E}_{11}$, $\mathcal{E}_{22}$, $\mathcal{E}_{33}$ are the strain components along three mutually orthogonal directions[27]. This results in a majority deviatoric solution, but does not prevent some degree of dilatation in the cuboid volumes[67]. Viscoelastic flow following earthquakes is driven by coseismic stress perturbations, as is the case for any postseismic dynamic models. In our inversion, we also implement a direction penalisation for the strain tensor of each cuboid to encourage the viscous strain to point in approximately the same direction as the coseismic stress changes[8].

**Checkerboard tests**. To investigate how well our combined near-field and far-field networks can resolve slip on the megathrust and strain within each cuboid, we perform a suite of checkerboard tests, illustrated in Supplementary Fig. 2. The input model is given in Supplementary Fig. 2a, where we use a checkerboard with ~220 km×~180 km rectangles of constant slip, synthetic input strain and displacements on the near-field and far-field networks. For the synthetic strains within the cuboids, we use the spatial distribution of the coseismic stress changes following the 2004 Sumatra–Andaman, the 2005 Nias–Simeulue and the 2007 Bengkulu earthquakes with these synthetic viscous strains scaled by the norm of the deviatoric coseismic stress changes. We first test both the near-field GPS network (SuGAr, Andaman GPS) and far-field (Thailand) GPS stations only, and the inverted results are shown in Supplementary Fig. 2b. In general, the slip pattern is relatively well-resolved in places near the GPS stations. The inverted strain distribution does not match the input model, with some additional strain occurring down dip of the 2005 rupture area and also in the northern tip of the mantle wedge. Due to there being no far-field sites for both the 2005 and 2007 rupture segments, the missed strain signals are mapped into the slip, causing overestimates on slip magnitude and a slight distortion of the spatial distribution of slip. We then test the near-field GPS network without the far-field GPS stations and tide gauges (Supplementary Fig. 2c). We note that the general pattern of the slip is relatively well-resolved excepting the middle section of the 2004 rupture area. However, the strain within the 2004 rupture region is missing, and there is additional artificial strain in the northern part of the 2004 down dip rupture area and the down dip region of the 2005 rupture region. Finally, we explored the combination of both the near-field and far-field networks (all GPS stations and tide gauges) in Supplementary Fig. 2d, which affords the best resolution among all these tests for both slip on the megathrust and strain within the cuboids. In general, the slip patterns are well-resolved, with the exception of the middle

section of the 2004 rupture due to a paucity of data within this region. In particular, the afterslip is best resolved at the shallow and deep part of the 2007 Bengkulu rupture area, the shallow and part of the deep part of the 2005 Nias–Simeulue, and the northern part of the 2004 Sumatra–Andaman rupture regions. For the strain, the best-resolved sections are the 2004 Sumatra–Andaman and the 2007 Bengkulu rupture segments. In conclusion, incorporation of the far-field tide gauges and the consequent increase in the total amount of available data can help to stabilise the inversion and improve the model resolution for each individual mechanism.

**Effective viscosity estimation**. We estimate the effective viscosity within each cuboid through

$$\eta_{\text{effective}} = \frac{\Delta\sigma(t)(t) + \sigma_0}{\Delta\dot{\varepsilon}(t) + \dot{\varepsilon}_0}, \tag{4}$$

where $\Delta\sigma$ is the norm of deviatoric stress evolution for each block, calculated from the accumulated viscous strain at each epoch estimated by the Kalman filter combined with the stress kernel associated with each cuboid (see refs. [22,27]), $\Delta\dot{\varepsilon}$ is the norm of deviatoric strain rate (as estimated by the Kalman filter), $\sigma_0$ and $\dot{\varepsilon}_0$ are the background stress and strain rate of the mantle wedge flow under the steady-state stage condition, respectively. For example, ref. [22] has demonstrated a static inversion for viscosity immediately following an earthquake, and in this study we expand it to capture the time-dependent features. It is impossible to directly evaluate the magnitude of stress and background strain rate in the mantle; thus, we explore a range of strain rates from $10^{-16}$ to $10^{-14}\,\text{s}^{-1}$, a range of steady-state viscosity from $10^{18}$ to $10^{22}\,\text{Pa}\,\text{s}$, and the background stress corresponds to the background strain rate multiplied by the steady-state viscosity. We loop through these parameters within the range, and calculate the misfit between Burgers rheology model and the effective viscosity. The minimum misfit provides the most likely combination of both background viscosity and strain rate that matches the trend of stress relaxation within the cuboid, and thus the corresponding time series of effective viscosity.

**Phase diagram**. We model the evolution of the stress and stress rate in a cuboid by using a simple spring-dashpot model. We consider the circumstance where the mechanical system is first at equilibrium under a constant stress $\tau_0$ but then subjected to a new stress $\tau_0 + \Delta\tau$ at time $t = 0$. The set of governing equation for the system is given by

$$\dot{\tau} = -G_{\text{M}}(\dot{\varepsilon}_{\text{M}} + \dot{\varepsilon}_{\text{K}})$$
$$\dot{\varepsilon}_{\text{M}} = \frac{\tau}{\eta_{\text{M}}} \tag{5}$$
$$\dot{\varepsilon}_{\text{K}} = \frac{\tau - G_{\text{K}}\varepsilon_{\text{K}}}{\eta_{\text{K}}}$$

where $G$, $\eta$ and $\varepsilon$ are the rigidity, viscosity and anelastic strain. The subscripts M and K are for the Maxwell and Kelvin materials, respectively. The evolution of stress is provided in closed form below for $t \geq 0$,

$$\tau\left(t; G_{\text{M}}, G_{\text{K}}, \eta_{\text{M}}, \eta_{\text{K}}, \tau_0, \Delta\tau\right) =$$

$$\frac{e^{\frac{t(\eta_{\text{K}}G_{\text{M}} + \eta_{\text{M}}(G_{\text{K}} + G_{\text{M}}))}{\eta_{\text{K}}\eta_{\text{M}}}}}{2\left(\eta_{\text{K}}^2 G_{\text{M}}^2 + 2\eta_{\text{K}}\eta_{\text{M}}(G_{\text{M}} - G_{\text{K}})G_{\text{M}} + \eta_{\text{M}}^2(G_{\text{K}} + G_{\text{M}})^2\right)}$$

$$\left\{\Delta\tau\left[e^{\frac{t(\eta_{\text{K}}G_{\text{M}} + \eta_{\text{M}}(G_{\text{K}} + G_{\text{M}}) + \Gamma)}{2\eta_{\text{K}}\eta_{\text{M}}}}\left(\eta_{\text{M}}^2(G_{\text{K}} + G_{\text{M}})^2\right.\right.\right.$$

$$+ \eta_{\text{K}}G_{\text{M}}(\eta_{\text{K}}G_{\text{M}} - \Gamma) - \eta_{\text{M}}(G_{\text{M}} - G_{\text{K}})(\Gamma - 2\eta_{\text{K}}G_{\text{M}}))$$

$$+ e^{\frac{t(\eta_{\text{K}}G_{\text{M}} + \eta_{\text{M}}(G_{\text{K}} + G_{\text{M}}) - \Gamma)}{2\eta_{\text{K}}\eta_{\text{M}}}}(\eta_{\text{M}}^2(G_{\text{K}} + G_{\text{M}})^2 \tag{6}$$

$$+ \eta_{\text{K}}G_{\text{M}}(\eta_{\text{K}}G_{\text{M}} + \Gamma) + \eta_{\text{M}}(G_{\text{M}} - G_{\text{K}})(2\eta_{\text{K}}G_{\text{M}} + \Gamma))\Big]$$

$$+ 2\tau_0\left[e^{\frac{t(\eta_{\text{M}}G_{\text{K}} + \eta_{\text{K}}G_{\text{M}} + \eta_{\text{M}}G_{\text{M}} - \Gamma)}{2\eta_{\text{K}}\eta_{\text{M}}}}\eta_{\text{M}}G_{\text{M}}\Gamma - e^{\frac{t(\eta_{\text{M}}G_{\text{K}} + \eta_{\text{K}}G_{\text{M}} + \eta_{\text{M}}G_{\text{M}} + \Gamma)}{2\eta_{\text{K}}\eta_{\text{M}}}}\eta_{\text{M}}G_{\text{M}}\Gamma\right.$$

$$+ e^{\frac{tG_{\text{M}}}{\eta_{\text{M}}} + \frac{t(G_{\text{K}} + G_{\text{M}})}{\eta_{\text{K}}}}(\eta_{\text{K}}^2 G_{\text{M}}^2 + 2\eta_{\text{K}}\eta_{\text{M}}(G_{\text{M}} - G_{\text{K}})G_{\text{M}}$$

$$\left.\left. + \eta_{\text{M}}^2(G_{\text{K}} + G_{\text{M}})^2)\right]\right\}$$

with

$$\Gamma = \sqrt{\eta_{\text{K}}^2 G_{\text{M}}^2 + 2\eta_{\text{K}}\eta_{\text{M}}(G_{\text{M}} - G_{\text{K}})G_{\text{M}} + \eta_{\text{M}}^2(G_{\text{K}} + G_{\text{M}})^2}. \tag{7}$$

The evolution of total anelastic strain rate $\dot{\varepsilon}^i = \dot{\varepsilon}_{\text{M}} + \dot{\varepsilon}_{\text{K}}$ is given by

$$\dot{\varepsilon}^i\left(t; G_{\text{M}}, G_{\text{K}}, \eta_{\text{M}}, \eta_{\text{K}}, \tau_0, \Delta\tau\right)$$

$$= \frac{e^{\frac{t(\eta_{\text{K}}G_{\text{M}} + \eta_{\text{M}}(G_{\text{K}} + G_{\text{M}}))}{\eta_{\text{K}}\eta_{\text{M}}}}}{4\eta_{\text{K}}\eta_{\text{M}}^2 G_{\text{M}}\left(\eta_{\text{K}}^2 G_{\text{M}}^2 + 2\eta_{\text{K}}\eta_{\text{M}}(G_{\text{M}} - G_{\text{K}})G_{\text{M}} + \eta_{\text{M}}^2(G_{\text{K}} + G_{\text{M}})^2\right)}\Big\{$$

$$\Delta\tau\eta_{\text{M}}\left[4e^{\frac{tG_{\text{M}}}{\eta_{\text{M}}} + \frac{t(G_{\text{K}} + G_{\text{M}})}{\eta_{\text{K}}}}(\eta_{\text{K}}G_{\text{M}} + \eta_{\text{M}}(G_{\text{K}} + G_{\text{M}}))\right.$$

$$- e^{\frac{t(\eta_{\text{K}}G_{\text{M}} + \eta_{\text{M}}(G_{\text{K}} + G_{\text{M}}) + \Gamma)}{2\eta_{\text{K}}\eta_{\text{M}}}}(\eta_{\text{K}}G_{\text{M}} + \eta_{\text{M}}(G_{\text{K}} + G_{\text{M}}) + \Gamma)$$

$$(\eta_{\text{M}}^2(G_{\text{K}} + G_{\text{M}})^2 + \eta_{\text{K}}G_{\text{M}}(\eta_{\text{K}}G_{\text{M}} - \Gamma)$$

$$- \eta_{\text{M}}(G_{\text{M}} - G_{\text{K}})(\Gamma - 2\eta_{\text{K}}G_{\text{M}}))$$

$$- e^{\frac{t(\eta_{\text{K}}G_{\text{M}} + \eta_{\text{M}}(G_{\text{K}} + G_{\text{M}}) - \Gamma)}{2\eta_{\text{K}}\eta_{\text{M}}}}(\eta_{\text{K}}G_{\text{M}} + \eta_{\text{M}}(G_{\text{K}} + G_{\text{M}}) - \Gamma)$$

$$(\eta_{\text{M}}^2(G_{\text{K}} + G_{\text{M}})^2 + \eta_{\text{K}}G_{\text{M}}(\eta_{\text{K}}G_{\text{M}} + \Gamma)$$

$$\left. + \eta_{\text{M}}(G_{\text{M}} - G_{\text{K}})(2\eta_{\text{K}}G_{\text{M}} + \Gamma))\right]$$

$$+ 2G_{\text{M}}\left[e^{\frac{t(\eta_{\text{K}}G_{\text{M}} + \eta_{\text{M}}(G_{\text{K}} + G_{\text{M}}) - \Gamma)}{2\eta_{\text{K}}\eta_{\text{M}}}}\right.$$

$$\Gamma(-\eta_{\text{K}}G_{\text{M}} - \eta_{\text{M}}(G_{\text{K}} + G_{\text{M}}) + \Gamma)\eta_{\text{M}}^2$$

$$+ e^{\frac{t(\eta_{\text{K}}G_{\text{M}} + \eta_{\text{M}}(G_{\text{K}} + G_{\text{M}}) + \Gamma)}{2\eta_{\text{K}}\eta_{\text{M}}}}\Gamma(\eta_{\text{K}}G_{\text{M}} + \eta_{\text{M}}(G_{\text{K}} + G_{\text{M}}) + \Gamma)\eta_{\text{M}}^2$$

$$+ 2e^{\frac{tG_{\text{M}}}{\eta_{\text{M}}} + \frac{t(G_{\text{K}} + G_{\text{M}})}{\eta_{\text{K}}}}\eta_{\text{K}}(\eta_{\text{K}}^2 G_{\text{M}}^2 + 2\eta_{\text{K}}\eta_{\text{M}}(G_{\text{M}} - G_{\text{K}})G_{\text{M}}$$

$$+ \eta_{\text{M}}^2(G_{\text{K}} + G_{\text{M}})^2)\eta_{\text{M}} \tag{8}$$

$$+ 2e^{\frac{tG_{\text{M}}}{\eta_{\text{M}}} + \frac{t(G_{\text{K}} + G_{\text{M}})}{\eta_{\text{K}}}}t(\eta_{\text{K}}G_{\text{M}} + \eta_{\text{M}}(G_{\text{K}} + G_{\text{M}}))$$

$$(\eta_{\text{K}}^2 G_{\text{M}}^2 + 2\eta_{\text{K}}\eta_{\text{M}}(G_{\text{M}} - G_{\text{K}})G_{\text{M}} + \eta_{\text{M}}^2(G_{\text{K}} + G_{\text{M}})^2)\Big]\tau_0$$

$$+ 2(\eta_{\text{K}}G_{\text{M}} + \eta_{\text{M}}(G_{\text{K}} + G_{\text{M}}))\left[\Delta\tau\eta_{\text{M}}\left(-2e^{\frac{tG_{\text{M}}}{\eta_{\text{M}}} + \frac{t(G_{\text{K}} + G_{\text{M}})}{\eta_{\text{K}}}}\right.\right.$$

$$(\eta_{\text{K}}^2 G_{\text{M}}^2 + 2\eta_{\text{K}}\eta_{\text{M}}(G_{\text{M}} - G_{\text{K}})G_{\text{M}} + \eta_{\text{M}}^2(G_{\text{K}} + G_{\text{M}})^2)$$

$$+ e^{\frac{t(\eta_{\text{K}}G_{\text{M}} + \eta_{\text{M}}(G_{\text{K}} + G_{\text{M}}) + \Gamma)}{2\eta_{\text{K}}\eta_{\text{M}}}}$$

$$(\eta_{\text{M}}^2(G_{\text{K}} + G_{\text{M}})^2 + \eta_{\text{K}}G_{\text{M}}(\eta_{\text{K}}G_{\text{M}} - \Gamma)$$

$$- \eta_{\text{M}}(G_{\text{M}} - G_{\text{K}})(\Gamma - 2\eta_{\text{K}}G_{\text{M}})) + e^{\frac{t(\eta_{\text{K}}G_{\text{M}} + \eta_{\text{M}}(G_{\text{K}} + G_{\text{M}}) - \Gamma)}{2\eta_{\text{K}}\eta_{\text{M}}}}$$

$$(\eta_{\text{M}}^2(G_{\text{K}} + G_{\text{M}})^2 + \eta_{\text{K}}G_{\text{M}}(\eta_{\text{K}}G_{\text{M}} + \Gamma)$$

$$+ \eta_{\text{M}}(G_{\text{M}} - G_{\text{K}})(2\eta_{\text{K}}G_{\text{M}} + \Delta)))$$

$$- 2G_{\text{M}}\left(-e^{\frac{t(2\eta_{\text{M}}G_{\text{K}} + \eta_{\text{K}}G_{\text{M}} + \eta_{\text{M}}G_{\text{M}} - \Delta)}{2\eta_{\text{K}}\eta_{\text{M}}}}\right.$$

$$\Gamma\eta_{\text{M}}^2 + e^{\frac{t(2\eta_{\text{M}}G_{\text{K}} + \eta_{\text{K}}G_{\text{M}} + \eta_{\text{M}}G_{\text{M}} + \Gamma)}{2\eta_{\text{K}}\eta_{\text{M}}}}\Gamma\eta_{\text{M}}^2$$

$$+ e^{\frac{tG_{\text{M}}}{\eta_{\text{M}}} + \frac{t(G_{\text{K}} + G_{\text{M}})}{\eta_{\text{K}}}}t(\eta_{\text{K}}^2 G_{\text{M}}^2 + 2\eta_{\text{K}}\eta_{\text{M}}(G_{\text{M}} - G_{\text{K}})G_{\text{M}}$$

$$+ \eta_{\text{M}}^2(G_{\text{K}} + G_{\text{M}}))\tau_0\Big]$$

We compared the analytic solution to a numerical integration using a fourth-order Runge–Kutta method, which showed agreement to numerical precision. We

model the phase diagram of Fig. 8 using Eqs. (5–8) by adjusting the physical parameters manually. The best-fit parameters for cuboid 2 are $\Delta\tau = 24.5$ kPa, $\tau_0 = 23$ kPa, $\eta_M = 2.45 \times 10^{18}$ Pa s, $\eta_K = 2.9 \times 10^{17}$ Pa s, $G_M = 30$ GPa and $G_K = 80$ GPa, respectively. For cuboid 3, they are $\Delta\tau = 30$ kPa, $\tau_0 = 31$ kPa, $\eta_M = 4.6 \times 10^{18}$ Pa s, $\eta_K = 3 \times 10^{17}$ Pa s, $G_M = 30$ GPa and $G_K = 100$ GPa.

**Data availability**. All geodetic data used in this study are available on request due to privacy or other restrictions.

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

## Acknowledgements

This research was supported by the National Research Foundation Singapore under its Singapore NRF Fellowship scheme (National Research Fellow Award Nos. NRF-NRFF2010-064 and NRF-NRFF2013-04), the Earth Observatory of Singapore via its funding from the National Research Foundation Singapore, and the Singapore Ministry of Education under the Research Centres of Excellence initiative. We thank Kelin Wang, Roland Bürgmann and an anonymous reviewer for their helpful comments. We are grateful to Prof. Ray J. Weldon for commenting on tide gauge data processing. We also would like to thank Jessica Murray and Ya-Ju Hsu for useful discussions about the ENIF. The modified ENIF code is available on request. Figures are produced by using Generic Mapping Tools (GMT)[68]. This is EOS paper number 142.

## Author contributions

Q.Q., E.M.H. and S.B. designed the study. Q.Q. conducted the study and wrote the manuscript with contributions from all other co-authors. L.F. prepared the GPS time series. J.D.P.M. derived the analytic solutions.

## Additional information

**Competing interests:** The authors declare no competing interest.

