## [Peer Review File · Nature Communications]

Reviewers' comments:

Reviewer #1 (Remarks to the Author):

The technical achievement as reported in this work is impressive. The authors handled a very large data set including both GPS and tide gauge measurements and were able to use an advanced new method to invert the data to estimate both afterslip distribution on the megathrust and strain rates in the mantle wedge. They further inferred "effective" viscosities of the mantle wedge from the strain rates. However, despite my admiration of the author team and their great scientific contribution over the past decade, it is difficult to offer the same compliments on the scientific accomplishment of this particular study. The study involves massive work but appears to have been done in haste. Given some time to develop a better thought out strategy, some critical technical and scientific issues could have been more adequately investigated. With the present manuscript, the reader cannot decide how much confidence to place in the reported results.

I would suggest that the authors redesign their work by initially focusing on a smaller region with good data coverage and thoroughly investigate model behaviour, resolution, and scientific implications. From the GPS data presented in the paper, it is obvious that the northern half of the study region is not affected by the two earthquakes in the south. Conversely, the southernmost area is little affected by the two great earthquakes in the north. There is a large "data gap" between latitudes 5 and 10. For all these reasons, it is neither necessary nor feasible to pursue a "master" inversion model for the entire length of the margin considered in this work, especially when applying a newly developed method that has yet to be thoroughly tested.

Much of the main conclusion is based on results from the "data gap" area mentioned above, illustrated by profile PP' in Fig. 2. The difficulty in separating contributions to surface deformation from afterslip and mantle wedge relaxation is well known, even with dense data coverage. With little data coverage, it is not possible to place much confidence in the afterslip and strain results shown in Fig. 1 (see resolution test for this region in Fig. E3). A few GPS sites in Thailand and site UMLH provide limited information for this region, but these sites exhibit the worst model fit (Fig. 2). In research, it is perfectly fine to propose author-preferred conceptual models with limited data (usually with the help of demonstrative forward modeling). In that situation, non-uniqueness is explicitly admitted and understood, and the proposed working hypothesis is based on consistency with the data as well as other theoretical and observational information. In this work, however, the results are determined by inversion, presented as "best-resolved" and "more robust", and said to be supported by resolution tests. The implied reliability of the results and conclusion thus becomes misleading.

The "sample of GPS stations" shown in Fig. E5 practically exclude all the stations that do not show good fit to observed cumulative displacements (Fig. 2), such as UMLH, LEWK, CARI, HUTB, PHKT, PTCT, PHUK, BANH, CPNT, and SAMP. This selection sacrifices objectivity. In fact, to understand why the few stations around the "data gap" area show poor fit in displacements, I tried to find their model and data time series in Fig. E5. But I could not find them.

The use of tide gauge data to constrain crustal deformation is a complex matter. The effort as briefly described in the paper does not seem adequate. Exactly how the tide gauge data improved the resolution (or not), which is difficult to tell by comparing Figs. E3b and E3d, needs to be better explored. Fig. E6 is not very useful; the models are mostly a sub-horizontal red line, regardless of the data. Why don't we see model variations similar to the GPS vertical components as shown in Fig. E5? If the contribution of the tide gauges is mainly to constrain the long-term trend, is the trend statistically significant for some of the sites with large fluctuations? It will be a good idea to work with some experts to do a better job with the tide gauge data before inverting them.

The estimation of effective viscosity is the core of the study as suggested by the title and abstract, but its derivation is very poorly described in the "Effective viscosity estimation" section. What is

meant by “we assume a certain range of strain rates?” Is it for the background strain rate? What range? What background viscosity is assumed? Is the linear Maxwell rheology for both σ_o and $\Delta\sigma$? If stress used in S3 is uniquely determined from strain rate using a linear function, can you expand S3 to let the viscosity be a function of strain rate only?

The application of a simple bi-viscous rheology as described in section “Effective viscosity of a simple b-viscous ...” does not seem to be correct. Equation S4 is shear strain solution for a uniform Burgers material with a specific boundary condition (constant shear stress load). Strain rates in the mantle wedge cannot be described by S4 alone, because they depend also on the interaction of the mantle wedge with the elastic overriding and subducting plates.

The use of oblique projection for Fig. 2 etc. is completely unnecessary. It saves little space, and there is no need to show these low-resolution results in very large figures. The mixed use of different map projections only creates difficulty for the reader. For example, it is quite an ordeal trying to compare time series results shown in Fig. E5 with displacement results in Fig. 2. I have to find station names from Fig. E1 and locate the stations in Fig. 2 which are shown with a different map projection. It is equally difficult to compare resolution tests (Fig. E3) with inversion results (Figs. 2, 3) because of different map projections. Even the simple task of translating the two lines PP' and QQ' from Fig. 2 to Fig. E3 proves to be very challenging; it is close to impossible to use coordinates because of the oblique projection in Fig. 2 and the lack of annotations for the south and east axes.

The rest of the comments go by line numbers.

65 onward. For a ten-year time window, the effect of megathrust locking cannot be ignored. If this effect is somehow removed from the data, it should be explained.

71. Reference 11 is not Hu et al.

76. One probably should say that viscoelastic relaxation of the oceanic lithosphere has limited influence in the short time frame considered in this work.

97. The afterslip Green's function is derived from an elastic model, presumably not the same as the inelastic model for mantle strain rates. In this case, is it correct to say that the “inversion approach intrinsically captures the coupling between afterslip and viscous flow”?

109. The spatial resolution for 2004 Sumatra-Andaman cannot be said to be good. Line 110 even implies that the afterslip estimate for the 2004 area is accurate. See “data gap” comment above.

122 onward. One needs to distinguish between material relaxation and system relaxation (see Wang et al., 2012, Nature). See comment about simple bi-viscous rheology above.

130 onward. The inversion results feature large spatial variations. The viscosity estimates shown in Fig. 3c are for specific areas, but the text here and the abstract seem to suggest that these are general results for the margin. Considering the spatial variations, the differences from previous models are not slight and cannot be due to “minor differences” in resolution.

144. Label Thailand in Fig. 3.

164-165. The statement is not right. The bi-viscous behaviour in the context of this work is also Newtonian. In the context of postseismic deformation, who would still think the asthenosphere is a viscous fluid?

240. Something is missing in this reference.

Review provided by Kelin Wang, April 2017.

Reviewer #2 (Remarks to the Author):

The authors synthesize a fascinating picture of the transient deformation around the active Sumatran plate boundary over the past 10 years and the evolution of upper mantle rheology. During this time, the region experienced several major earthquakes followed by periods of transient deformation driven by a combination of afterslip and uppermost mantle relaxation. This study affords an opportunity to gain a phenomenological picture of how effective viscosity varies with time and to test our conceptual and laboratory-based models on how rheology depends on proxy parameters (e.g., Kelvin and Maxwell viscosities of a Burgers body) and physical conditions such as temperature, water content, and grain size, as well as evolving stress. Their results carry important implications for the physical factors that shape mantle viscosity and, by extension, the regional seismic hazards through the evolving crustal stress field generated by the mantle flow.

The authors use a wealth of GPS data, combined with tide gauge data, that represents more than 10 years of measurements from the Sugar GPS network. They have implemented a novel method (based on Barbot et al., 2016) of simultaneously inferring temporally varying afterslip and 'distributed strain in finite volumes', which translates into effective viscosity of the mantle wedge along the Sumatran megathrust. This technical achievement alone is remarkable because the few previous studies that have explored joint afterslip / viscoelastic relaxation modeling in any tectonic environment have relied on ad-hoc parameterizations of temporally varying afterslip (at least for kinematic afterslip models), combined with a large number of forward model computations to infer the optimal viscosity structure. The direct inversion approach developed and adapted by the group in this study is a more appealing and powerful way of modeling the deformation data with these processes.

I strongly recommend publication in Nature, with minor revisions

Specific comments

Effective viscosities are often interpreted in terms of a non-linear flow law, with dependence on temperature, water content, grain size, and stress exponent. The authors' numerical model contains all of the information necessary to compare with such a flow law. The comparison with the predictions of a Burgers body are already a step in this direction, but perhaps more insight is possible by a brief comparison of inferred effective viscosity (at a chosen location in the mantle wedge) with the prediction of a non-linear flow law. Stress in the ductile mantle is decreasing with time, which already will lead to a temporally increasing effective viscosity, but is this sufficient to capture what is shown in, say, Figure 3c? I realize that a combined transient rheology and long-term rheology flow law (e.g., eqn 4 of Freed et al., 2012) might be necessary for such a comparison.

Figure 2a. It is somewhat surprising that, given the number of free parameters, the GPS displacement vectors are not fit better in this figure. It seems that additional afterslip would take care of some of the remaining mismatch, but somehow the inversion did not prefer this.

line 40 of the supplemental text. In the web address change avso to aviso

'Afterslip penalisation and smoothing regularization' section of supplemental text. A few more details would help here. How were the after slip parameters with the given constraints determined -- MCMC, simulated annealing, etc.? Would the amount of smoothing affect the misfit that still remains in Figure 2a alluded to above?

Reference

Freed, A. M., G. Hirth, and M. D. Behn, Using short-term postseismic displacements to infer the ambient deformation conditions of the upper mantle, *J. Geophys. Res.*, 117, B01,409, doi:10.1029/2011JB008,562, 2012.

Response to Reviewers' comments:

Reviewer #1 (Remarks to the Author):

The technical achievement as reported in this work is impressive. The authors handled a very large data set including both GPS and tide gauge measurements and were able to use an advanced new method to invert the data to estimate both afterslip distribution on the megathrust and strain rates in the mantle wedge. They further inferred "effective" viscosities of the mantle wedge from the strain rates. However, despite my admiration of the author team and their great scientific contribution over the past decade, it is difficult to offer the same compliments on the scientific accomplishment of this particular study. The study involves massive work but appears to have been done in haste. Given some time to develop a better thought out strategy, some critical technical and scientific issues could have been more adequately investigated. With the present manuscript, the reader cannot decide how much confidence to place in the reported results.

We are grateful for this detailed review and careful reading of our paper.

Our manuscript was originally submitted to Nature, and then transferred to Nature Communication via Nature Geoscience. Therefore, within the length constraints provided by the Nature format, we were unable to put more technique details in front of readers, opting instead to focus on the key results of our study. Now that we are working with the Nature Communications format we are able to include a detailed methods section in the main content and weave it more cohesively into the narrative. Thus, we anticipate the detailed methods section will help the reader to better understand the framework and main concepts of the novel approach highlighted in this manuscript.

We have taken great pains to combine, select, process and to deal with the data uncertainties of the multiple geodetic time series (e.g., GPS and tide gauges) to help us to constrain the postseismic deformation. We acknowledge that these data sets are not perfect and have variable spatial coverage, which in turn results in spatial variabilities in the resolving power on the megathrust and in the mantle wedge. However, these time series (the complete data sets but not including GRACE data) represent a decade-long series of surface displacements, with the temporal resolution and longevity to constrain the evolution of the postseismic relaxation mechanisms. In our manuscript, we focus on the time-evolution of the effective viscosity constrained by these time series, rather than attempting to interpret spatial heterogeneities. Thus we can place reasonable confidence on the estimated time evolution of the effective viscosity. We have also added an additional sentence at line 58 to highlight the high-resolution temporal constraints on the postseismic mechanisms, repeated below:

“The resulting time series of displacements span the time period from early 2005 through to 2014, providing constraints for the temporal evolution of the rheological parameters of the various postseismic relaxation mechanisms.”

To better understand the mechanical coupling between afterslip and viscous flow, we have included the surface displacement predictions from both afterslip and viscous flow for stations down-dip of the 2007 Bengkulu rupture in Figure 5. We further conducted forward viscous modelling utilising the rheological structure presented by *Panet et al. (2010)* for the 2004 Sumatra-Andaman earthquake, assuming the structure at these two segments are similar. We used the coseismic slip distribution from *Konca et al. (2008)* as the input to drive VISCO1D models to calculate the surface displacements, which are also summarised in Figure 5. We note that the temporal curvatures of surface displacement contributed by viscous flow are well captured by our inversion as compared with the dynamic forward model. For some components of stations (e.g., east, north, up), the amplitude is also captured well. We have included sentences in the main content at lines 119-135, replicated below.

“Our approach can, in principle, disentangle the contributions of afterslip and viscoelastic flow, provided enough data are available. To assess the potential and limitations of this experimental setting, we compare our kinematic results with dynamic simulations at four stations (PSKI, LNNG, MKMK, and LAIS) in the relatively well-resolved Bengkulu area. We conducted two stress-driven simulations assuming a gravitational, vertically stratified viscoelastic Earth using VISCO1D^{27,28} assuming the rheological structure adopted by ref. 17 for the 2004 Sumatra-Andaman earthquake (Burgers body, Simulation 1), and the same rheological parameters from ref. 17 but without the transient creep component (Maxwell body, Simulation 2). The initial coseismic stress changes are calculated from ref. 8. We show the predicted time evolution of the surface displacement, including contributions from both afterslip and viscoelastic flow following the 2007 Bengkulu earthquake (Fig. 5). Our modelled displacements sit half way between the Maxwell and the Burgers models, indicating some consistency with previous models of the region. The small differences in inferred viscosity may be due to our modelling assumptions. In particular, our inversion allows for the possibility of arbitrary three-dimensional heterogeneities, whilst previous approaches were restricted to vertical stratification and did not give afterslip a consistent treatment.”

We have also compared our transient and long-term viscosity estimates with the effective viscosities reported in published rheological models from the Sumatran subduction zone (*Han et al. 2008; Panet et al. 2010; Pollitz et al. 2008; Hoechner et al. 2011; Hu and Wang. 2012; Broerse et al. 2015*). We further quantitatively compared the characteristic time scale for the transient behaviour between the published ones and ours. We found that they agree with each other fairly well (Table 1). We have highlighted this in the main content at lines 174-184, repeated below.

“We note the reasonable agreement between the effective viscosity time-series and the simplified model of eq. (1) (e.g., Fig. 6c). Our estimated transient and background viscosities are on the order of $\sim 10^{17}$ Pa s and $\sim 10^{19}$ Pa s respectively, with a characteristic time scale of 0.21 ± 0.05 yr for the transient. The time scale of the transient is commensurate with that found in other tectonic settings (e.g., ref. 29) and our effective viscosity estimates from our example cuboids are also in agreement with several published Burgers rheology models for the Sumatran subduction zone^{11,14–17,19,20} (Table 1) despite the different modelling assumptions and techniques. These results imply that the time evolution of the viscosity is reasonably well recovered using our approach. The remaining differences can be attributed to the longer time series used in this study, and the fact that we do not constrain *a priori* the lateral and temporal variations in viscosity.”

I would suggest that the authors redesign their work by initially focusing on a smaller region with good data coverage and thoroughly investigate model behaviour, resolution, and scientific implications. From the GPS data presented in the paper, it is obvious that the northern half of the study region is not affected by the two earthquakes in the south. Conversely, the southernmost area is little affected by the two great earthquakes in the north. There is a large “data gap” between latitudes 5 and 10. For all these reasons, it is neither necessary nor feasible to pursue a “master” inversion model for the entire length of the margin considered in this work, especially when applying a newly developed method that has yet to be thoroughly tested.

Thank you for your suggestion to break the whole study into to individual specific studies. Individual studies have been, and will be, done by our group members for individual segments, with focus of those to better understand the relative contribution of afterslip and viscoelastic flow, and spatial and temporal evolution of the afterslip on the megathrust. In this study, our goal is to get an overall view of the rheology of the Sumatran subduction zone by using decade long multiple geodetic time series.

We agree that there is a data gap in the middle segment of the 2004 Sumatra-Andaman rupture area. Unfortunately, there are no data available for this region to help provide constraints for distribution of afterslip and viscous flow mechanisms at this segment. Missing data will likely affect the amplitude of the estimated afterslip and viscous strain. In fact, it is extremely difficult to accurately constrain the amplitudes of any individual mechanism even with a far denser set of geodetic measurements, and other published techniques. This is because the natural trade-offs of these multiple mechanisms co-contribute to the surface displacement over the postseismic relaxation process, in many cases generating similar spatial deformation patterns across large portions of the rupture zone. Therefore, we are focusing on the time evolution of the estimated effective viscosity constrained by the best available high-resolution decade long time series. It is important to note that by incorporating the far-field tide gauge time series the model resolution for the viscous strain is significantly enhanced (see Methods section and Supplementary Fig. 2 for information on checkerboard tests), helping to stabilize the inversion results and to gain a reasonable spatial pattern of the afterslip and viscous strain in the mantle wedge.

Much of the main conclusion is based on results from the “data gap” area mentioned above, illustrated by profile PP’ in Fig. 2. The difficulty in separating contributions to surface deformation from afterslip and mantle wedge relaxation is well known, even with dense data coverage. With little data coverage, it is not possible to place much confidence in the afterslip and strain results shown in Fig. 1 (see resolution test for this region in Fig. E3).

We have discussed the issue of the data gap in the previous paragraph. Areas of reduced data coverage will bias the amplitude estimates for both afterslip and viscous strain. However, the time series provide robust constraints on the time-evolution of the effective viscosity. Additionally, the spatial resolution of the viscous strain is greatly enhanced with the incorporation of the far-field tide-gauge time series (see the colour changes at these cuboids downdip of the 2004 Sumatra-Andaman rupture in Supplementary Fig. 2c, d).

A few GPS sites in Thailand and site UMLH provide limited information for this region, but these sites exhibit the worst model fit (Fig. 2). In research, it is perfectly fine to propose author-preferred conceptual models with limited data (usually with the help of demonstrative forward modeling). In that situation, non-uniqueness is explicitly admitted and understood, and the proposed working hypothesis is based on consistency with the data as well as other theoretical and observational information. In this work, however, the results are determined by inversion, presented as “best-resolved” and “more robust”, and said to be supported by resolution tests. The implied reliability of the results and conclusion thus becomes misleading.

The published GPS sites in Thailand are collected from *Broerse et al. (2015)*. They are not all continuous stations. In fact, most of them are campaign sites. Thus, they have relatively large uncertainties when compared to the continuous stations. Another significant issue that must be considered is that different stations cover different time periods, with different starting times. Many stations were installed after the 2004 earthquake. Thus, it is not possible to know the postseismic displacements accumulated immediately after the 2004 rupture until the stations were installed, resulting in a shift between the model and the data. The same phenomenon also affects station UMLH. Due to a lack of data before the 2004 earthquake, the interseismic velocity is poorly constrained (*Feng et al. 2015*), resulting in a large uncertainty through fitting the time series (*Feng et al. 2015*). As the large uncertainties are used in the covariance matrix for the Kalman filter, it forces the filter to put less weight on fitting the time series measurement for these stations during the Kalman filter updating step. Together, the model fit for these sites is correspondingly weaker.

Forward models based on the proposed assumptions are much easier, because the model will straightforwardly predict the surface displacement from the assumed mechanisms. Geodetic

inversion is more difficult, because it is purely data driven, highly dependent on the geodetic network, where and how much data are available. Thus, with the geodetic network coverage limitations we need to better understand the spatial variability of the model resolution. We conducted a number of checkerboard tests to investigate the contribution of each geodetic dataset. These tests (Supplementary Fig. 2) are very helpful in helping us to understand which parts of the megathrust and mantle wedge are well resolved, and can thus help us to build confidence for the estimated time evolution of the effective viscosity.

The “sample of GPS stations” shown in Fig. E5 practically exclude all the stations that do not show good fit to observed cumulative displacements (Fig. 2), such as UMLH, LEWK, CARI, HUTB, PHKT, PTCT, PHUK, BANH, CPNT, and SAMP. This selection sacrifices objectivity. In fact, to understand why the few stations around the “data gap” area show poor fit in displacements, I tried to find their model and data time series in Fig. E5. But I could not find them.

These stations are located in Thailand, Andaman Islands and northern Sumatra. Except the SAMP station, the remaining stations started to record the postseismic displacement after the 2004 Sumatra-Andaman earthquake. As we mentioned in the previous paragraph, the early (and fast) accumulated postseismic displacements were missed at these stations. Thus, the model and the data have an offset. For the SAMP station, the model fits with the data within the data uncertainty (Fig. 4), which is correspondingly large. We have large uncertainty estimations for stations e.g., UMLH and RNGT. We have included another Supplementary Fig. 4 to show all the model fits with GPS time series, and the model fits with all the tide gauge time series in Supplementary Fig. 5.

The use of tide gauge data to constrain crustal deformation is a complex matter. The effort as briefly described in the paper does not seem adequate. Exactly how the tide gauge data improved the resolution (or not), which is difficult to tell by comparing Figs. E3b and E3d, needs to be better explored. Fig. E6 is not very useful; the models are mostly a sub-horizontal red line, regardless of the data. Why don't we see model variations similar to the GPS vertical components as shown in Fig. E5? If the contribution of the tide gauges is mainly to constrain the long-term trend, is the trend statistically significant for some of the sites with large fluctuations? It will be a good idea to work with some experts to do a better job with the tide gauge data before inverting them.

We have actually made a great deal of effort to tackle the processing issue. We have discussed the processing of tide gauge data at length with expert Prof. Ray J. Weldon, Professor, at the Department of Geological Sciences, University of Oregon. We have used his simple but robust way to remove the seasonal noise (*Burgette et al. 2009*) from both the tide-gauge data and Aviso satellite altimetry time series. We believe this is sufficient to deal with the seasonal signals.

Actually, we see clear long-term linear trend deviation after the megathrust earthquakes at the Sumatran subduction zone e.g., the 2004 Mw 9.2 Sumatran-Andaman, the 2005 Mw 8.5 Nias-

Simeulue and the 2007 Bengkulu earthquakes, but they were not clear in the figure we had previously submitted due to our plotting of data only after 2004. We have included Supplementary Fig. 5 to show the clear changes. The resulting land-height change time series shown in Supplementary Fig. 5 are still noisy with a significant large uncertainty, due to the low spatial resolution of the Aviso grid. Thus, the local complexity of the bathymetry effect that was captured by the tide gauges was not recorded on the neighbouring Aviso grid. Such an effect might be mapped into the land-height time series, resulting in some fluctuations. However, they do show a land-subsidence trend at most of the stations that were affected by the earthquakes, providing tight time evolution constraints on the postseismic mechanisms. We have included a paragraph at lines 301-323.

“We minimised the uncertainty by carefully removing the variable seasonal signals, and by finding the collocated tide gauge and Aviso grids pairs. However, there are a number of uncertainties that are still contained within the resultant land-height time series due to the low spatial resolution of the Aviso grids. For example, the local complexities of shallow bathymetric effects and manmade land-height signals that would be captured by the tide gauges are absent in the coarse altimetry Aviso grids. These localised signals would then be mapped into the land-height time series, potentially producing abnormalities in the land-height variation when compared to neighbouring tide gauges that were less affected by the local signals. For example, the high rate of subsidence at tide gauge BPAK-Bang Pakong (Fig. 2) is comparable or even larger than that of tide gauge TAPN- Ko Taphao Noi (Fig. 2), which was closer to the 2004 Sumatra-Andaman earthquake, and strongly affected by the postseismic deformation process (Supplementary Fig. 5). This abnormally high rate of subsidence likely contains contamination from human-induced subsidence at Bangkok, Thailand (see refs 55,56), approximately 50 km away. Tide gauge RAYG-Rayong (see Fig. 2) shows strong variations preceding the 2004 Mw 9.2 Sumatra-Andaman earthquake (Supplementary Fig. 5). Similar abnormal signals were also found at tide gauges east of Malaysia, which were less affected by the 2004 rupture. To completely and accurately deal with these abnormal signals and uncertainties presents a substantial challenge. However, the tide gauge data do clearly show changes in their linear trend after the great megathrust events (vertical dashed lines in Supplementary Fig. 5), which provides a strong constraint on the direction of land movement, if not its magnitude, affording a crucial constraint for the mechanisms of the postseismic relaxation processes (see red profiles in Supplementary Fig. 5).”

The tide gauges play an important role in terms of constraining the postseismic mechanisms. As the additional time series of these tide gauges, it increase the data space and stabilize the inversion process, and come with reasonable spatial pattern of the both afterslip and viscous strain (Supplementary Fig. 2). The most import role of the tide gauge is that they robustly show subsidence, providing a key constraint for the nature of the rheology.

The estimation of effective viscosity is the core of the study as suggested by the title and abstract, but its derivation is very poorly described in the “Effective viscosity estimation” section. What is meant by “we assume a certain range of strain rates?” Is it for the background strain rate? What range? What background viscosity is assumed? Is the linear Maxwell rheology for both σ_0 and

delta sigma? If stress used in S3 is uniquely determined from strain rate using a linear function, can you expand S3 to let the viscosity be a function of strain rate only?

We have clarified the methodology in the main text and the "Effective viscosity estimation" section of the Methods. We have included these in lines 440-443:

“It is impossible to directly evaluate the magnitude of stress and background strain rate in the mantle; thus, we explore a range of strain rates from 10^{-16} to 10^{-14} s⁻¹, a range of steady-state viscosity from 10^{18} to 10^{22} Pa s, and the background stress corresponds to the background strain rate multiplied by the steady-state viscosity.”

The linear Maxwell rheology is for sigma_o only. Sigma_o represents the background stress at steady state. We have made it clearer at lines 436-437:

“,... background stress and strain rate of the mantle wedge flow under the steady-state condition respectively.”

The stress (delta sigma (t)) is not uniquely determined from the strain rate; instead, it is calculated by using the accumulated strain at each time epoch and the stress kernel associated with the unit strain of each cuboid. We have made it clear at lines 433-435, and also added the citation for calculating the stress kernel for unit strain of each cuboid:

“..., calculated from the accumulated strain at each epoch estimated by the Kalman filter combined with the stress kernel associated with each cuboid (see ref. 1,13),...”

The application of a simple bi-viscous rheology as described in section “Effective viscosity of a simple b-viscous ...” does not seem to be correct. Equation S4 is shear strain solution for a uniform Burgers material with a specific boundary condition (constant shear stress load). Strain rates in the mantle wedge cannot be described by S4 alone, because they depend also on the interaction of the mantle wedge with the elastic overriding and subducting plates.

We are grateful for your comments on the stress and strain rate relation that might apply in the mantle wedge. This is absolutely right. The estimated stresses for the cuboids are indeed decaying with time. We utilise equation (1) as to capture the time evolution of the effective viscosity and

extract the initial and long-term viscosities. We have updated our manuscript accordingly at lines 162-167.

“To extract the transient and steady-state viscosities, we fit our time series of effective viscosities with the exponential function

$$\eta_{\text{eff}}(t) = \frac{\eta_K \eta_M}{\eta_M e^{\frac{t}{\tau_K}} + \eta_K} . \quad (1)$$

The choice of the functional is for convenience only and does not imply any particular underlying physical mechanism.”

The use of oblique projection for Fig. 2 etc. is completely unnecessary. It saves little space, and there is no need to show these low-resolution results in very large figures. The mixed use of different map projections only creates difficulty for the reader. For example, it is quite an ordeal trying to compare time series results shown in Fig. E5 with displacement results in Fig. 2. I have to find station names from Fig. E1 and locate the stations in Fig. 2 which are shown with a different map projection. It is equally difficult to compare resolution tests (Fig. E3) with inversion results (Figs. 2, 3) because of different map projections. Even the simple task of translating the two lines PP' and QQ' from Fig. 2 to Fig. E3 proves to be very challenging; it is close to impossible to use coordinates because of the oblique projection in Fig. 2 and the lack of annotations for the south and east axes.

Along the latitude direction, our study area covers a vast area of length ~3000 – 4000 km extending from southern tip of Sumatra to northern tip of Andaman Islands. While in the longitude direction, it is less than half of the distance. Thus, a different choice of projection would make the figures far less reader friendly. We think the projection used here is ideal for presenting our model results of both afterslip and viscous strain, and a good arrangement on showing the example cross-sections along the PP' and QQ' locations. We added the profiles PP' and QQ' to Fig. 2, but we respectfully request to keep the projection of Fig. 3 as is.

The rest of the comments go by line numbers.

65 onward. For a ten-year time window, the effect of megathrust locking cannot be ignored. If this effect is somehow removed from the data, it should be explained.

We have clarified it in the main text at lines 61-62. In addition, we have included a detailed methods section (GPS subsection) that covers the geodetic data processing to derive the postseismic time series.

“We model both the near- and far-field postseismic time series (see Methods) with localised slip on the fault and distributed strain in finite volumes^{1,12,13} ...”

71. Reference 11 is not Hu et al.

Fixed.

76. One probably should say that viscoelastic relaxation of the oceanic lithosphere has limited influence in the short time frame considered in this work.

Done.

97. The afterslip Green’s function is derived from an elastic model, presumably not the same as the inelastic model for mantle strain rates. In this case, is it correct to say that the “inversion approach intrinsically captures the coupling between afterslip and viscous flow”?

We assume that the stress interactions can be modelled within the framework of infinitesimal strain. In this case, the stress in the medium is a linear combination of the effects of afterslip and viscoelastic flow. We use the same earth model for afterslip and viscoelastic flow. With this assumption, it is entirely correct to say that the inversion approach intrinsically captures the coupling between the two source mechanisms.

109. The spatial resolution for 2004 Sumatra-Andaman cannot be said to be good. Line 110 even implies that the afterslip estimate for the 2004 area is accurate. See “data gap” comment above.

We agree. We modified the language accordingly at lines 139-143:

“.....As suggested from the resolution experimental tests (Supplementary Fig. 2), we are not able to estimate viscous strain and afterslip in the 2005 Nias–Simeulue rupture segment as accurately as the 2007 Bengkulu segment, due to the paucity of data directly downdip of the 2005 rupture above the viscous cuboids.”

122 onward. One needs to distinguish between material relaxation and system relaxation (see Wang et al., 2012, Nature). See comment about simple bi-viscous rheology above.

We agree. Our estimates shown in Fig. 6 are for effective *in situ* viscosity, which is not necessarily an intrinsic material property. Eq. (1) captures the temporal evolution of the viscosity and is convenient

to identify the transient and steady-state values of effective viscosity. The choice of the functional form of Eq. (1) is for convenience only and does not reflect any particular underlying mechanism. We note that the instantaneous effective viscosity immediately after the mainshock may be a local intrinsic property, as it is not yet influenced by the overall response of the system.

130 onward. The inversion results feature large spatial variations. The viscosity estimates shown in Fig. 3c are for specific areas, but the text here and the abstract seem to suggest that these are general results for the margin. Considering the spatial variations, the differences from previous models are not slight and cannot be due to “minor differences” in resolution.

We have spatially variable resolutions due to different data coverage. Although the spatial coverage of the data is not dense, we have made considerable efforts to combine the available datasets to constrain the postseismic mechanisms. To help us better understand which cuboids are well resolved, we have conducted the resolution tests through checkerboards. Our interpretation is based on the area that is well resolved based on the combined near- and far-field networks. Figure 6c gives two examples to show the time evolution of the effective viscosity. The spatial distribution of the fitted transient (Kelvin) and steady-state (Maxwell) viscosity of the well resolved cuboids are shown in Figure 6a, and b respectively. The unresolved cuboids are shaded with a light yellow colour. Thus, we interpret the effective viscosity estimation based on the first-order magnitude found in these well-resolved locations. The absolute magnitude of the viscosity depends on several factors e.g., the spatial and temporal resolution of the data, the technique being used for the viscous correction, the natural trade-offs of the afterslip and viscous flow, uncertainty of the data sets etc. These factors vary across different research studies, resulting in some variations among the published results. In our updated version, we have compared our transient and long-term viscosity estimates with the effective viscosities reported in published rheological models from the Sumatran subduction zone (*Han et al. 2008; Panet et al. 2010; Pollitz et al. 2008; Hoechner et al. 2011; Hu and Wang. 2012; Broerse et al. 2015*). We further quantitatively compared the characteristic time scale for the transient behaviour between the published ones and ours. We found that they agree with each other fairly well (Table 1). We have highlighted this in the main content at lines 174-184, repeated below.

“We note the reasonable agreement between the effective viscosity time-series and the simplified model of eq. (1) (e.g., Fig. 6c). Our estimated transient and background viscosities are on the order of $\sim 10^{17}$ Pa s and $\sim 10^{19}$ Pa s respectively, with a characteristic time scale of 0.21 ± 0.05 yr for the transient. The time scale of the transient is commensurate with that found in other tectonic settings (e.g., ref. 29) and our effective viscosity estimates from our example cuboids are also in agreement with several published Burgers rheology models for the Sumatran subduction zone^{11,14–17,19,20} (Table 1) despite the different modelling assumptions and techniques. These results imply that the time evolution of the viscosity is reasonably well recovered using our approach. The remaining differences can be attributed to the longer time series used in this study, and the fact that we do not constrain *a priori* the lateral and temporal variations in viscosity.”

144. Label Thailand in Fig. 3.

Done.

164-165. The statement is not right. The bi-viscous behaviour in the context of this work is also Newtonian. In the context of postseismic deformation, who would still think the asthenosphere is a viscous fluid?

We agree. We changed to the following statement at lines 229-231:

"Our study highlights that at least two creep mechanisms accommodate the viscoelastic flow of the asthenospheric upper mantle, which can be captured by a linear bi-viscous rheology, at least to first order."

240. Something is missing in this reference.

Fixed.

Review provided by Kelin Wang, April 2017.

We thank Dr. Wang for his thoughtful feedback that allowed us to greatly improve the manuscript.

Reviewer #2 (Remarks to the Author):

The authors synthesize a fascinating picture of the transient deformation around the active Sumatran plate boundary over the past 10 years and the evolution of upper mantle rheology. During this time, the region experienced several major earthquakes followed by periods of transient deformation driven by a combination of afterslip and uppermost mantle relaxation. This study affords an opportunity to gain a phenomenological picture of how effective viscosity varies with time and to test our conceptual and laboratory-based models on how rheology depends on proxy parameters (e.g., Kelvin and Maxwell viscosities of a Burgers body) and physical conditions such as temperature, water content, and grain size, as well as evolving stress. Their results carry important implications for the physical factors that shape mantle viscosity and, by extension, the regional seismic hazards through the evolving crustal stress field generated by the mantle flow.

The authors use a wealth of GPS data, combined with tide gauge data, that represents more than 10 years of measurements from the Sugar GPS network. They have implemented a novel method (based on Barbot et al., 2016) of simultaneously inferring temporally varying afterslip and 'distributed strain in finite volumes', which translates into effective viscosity of the mantle wedge along the Sumatran megathrust. This technical achievement alone is remarkable because the few previous studies that have explored joint afterslip / viscoelastic relaxation modeling in any tectonic environment have relied on ad-hoc parameterizations of temporally varying afterslip (at least for kinematic afterslip models), combined with a large number of forward model computations to infer the optimal viscosity structure. The direct inversion approach developed and adapted by the group in this study is a more appealing and powerful way of modeling the deformation data with these processes.

I strongly recommend publication in Nature, with minor revisions

We are grateful for this positive and helpful review.

Specific comments

Effective viscosities are often interpreted in terms of a non-linear flow law, with dependence on temperature, water content, grain size, and stress exponent. The authors' numerical model contains all of the information necessary to compare with such a flow law. The comparison with the predictions of a Burgers body are already a step in this direction, but

perhaps more insight is possible by a brief comparison of inferred effective viscosity (at a chosen location in the mantle wedge) with the prediction of a non-linear flow law. Stress in the ductile mantle is decreasing with time, which already will lead to a temporally increasing effective viscosity, but is this sufficient to capture what is shown in, say, Figure 3c? I realize that a combined transient rheology and long-term rheology flow law (e.g., eqn 4 of Freed et al., 2012) might be necessary for such a comparison.

We have added Figure 7 to highlight the in situ stress/strain-rate relationship at two cuboids near the 2004 and the 2007 earthquakes. We found that the data is well captured by the theoretical response of a Burgers material, which we detail in the Methods section. We could not explain the same dataset with powerlaw flow. We added the following paragraph at lines 210-225.

" The stress-strain rate phase diagram in log-log space (Fig. 8) reveals two distinct domains that we interpret as the signatures of transient creep and steady-state creep. Following both the 2004 Mw 9.2 Sumatra-Andaman earthquake and the 2007 Mw 8.4 Bengkulu earthquake, the transient domain reveals a nonlinear relationship that dominates the deformation for about two years, followed by a contrasting steady state creep which obeys a near linear stress-strain rate relationship. These results are compatible with a combination of steady-state creep and transient creep, which may be attributed to background diffusion creep plus temporary motion of the soft slip system of the olivine crystals². The stress/strain-rate behaviour at cuboids 2 and 3 is well captured by the theoretical response of a Burgers material to a stress perturbation added to a background load (Fig. 8; see Methods) and we could not explain these data with a power-law rheology with a power exponent of 2 or greater. We conclude that a Burgers rheology involving linear work-hardening transient creep and steady-state creep is a realistic description of mantle rock rheology within a time scale of days to years. We anticipate that future geodetic data sets, with greater spatial and temporal resolution both inland and offshore, will help us to gain more insight into the rheological structure of the region."

Figure 2a. It is somewhat surprising that, given the number of free parameters, the GPS displacement vectors are not fit better in this figure. It seems that additional afterslip would take care of some of the remaining mismatch, but somehow the inversion did not prefer this.

We have addressed a similar question for reviewer 1:

These stations are located in Thailand, Andaman Islands and northern Sumatra. Except the SAMP station, the remaining stations started to record the postseismic displacement after the 2004 Sumatra-Andaman earthquake. As we mentioned in previous paragraph, the early (and fast) accumulated postseismic displacements were missed at these stations. Thus, the model and the data have an offset. For the SAMP station, the model fits with the data within the data uncertainty (Fig.

4), which is correspondingly large. We have large uncertainty estimations for stations e.g., UMLH and RNGT. As the large uncertainties are used in the covariance matrix for the Kalman filter, it forces the filter to put less weight on fitting the time series measurement for these stations during the Kalman filter updating step. Together, the model fit for these sites is correspondingly weaker.

line 40 of the supplemental text. In the web address change avso to aviso

Fixed.

'Afterslip penalisation and smoothing regularization' section of supplemental text. A few more details would help here. How were the after slip parameters with the given constraints determined -- MCMC, simulated annealing, etc.? Would the amount of smoothing affect the misfit that still remains in Figure 2a alluded to above?

The spatial and temporal smoothing hyper-parameters are estimated together with the model parameters for each model run, which is also one of the advantages of the Extended Network Inversion Filter technique (*Segall and Matthews, 1997; McGuire and Segall, 2003*). The Kalman filter is a Bayesian filter, and these parameters are predicted through the transition matrix and updated by observations at each step. With more and more information available through time, these parameters are estimated better and better. The filter starts with an initial value, then adjusts it within the specified uncertainty to better fit the measurement. There is a trade-off between the spatial and temporal smoothing hyper-parameters. To better handle this problem, we first tried a few initial runs to determine a reasonable spatial smoothing hyper-parameter that can predict a smooth slip and reasonable fits to the data. Then we use this value with a small covariance that allows this value to be changed by the filter slightly over the Kalman update step, while using a large covariance for the temporal smoothing hyper-parameter that allows a large range of adjustment when necessary during the updating step. We then determine the initial guess for the temporal smoothing hyper-parameter, through recursively running the filter with the previous estimates as the current input, until we find the current hyper-parameter estimates and previous ones can produce similar model results (*McGuire and Segall, 2003*). We have included this paragraph in the main content at lines 365 -383:

“The spatial and temporal smoothing hyper-parameters are estimated together with the model parameters for each model run, which is also one of the advantages of the Extended Network Inversion Filter^{22,23} technique. The Kalman filter is a Bayesian filter, where the parameters are predicted through a transition matrix and updated with the measurement from the current step. Then we step through to the next time epoch. As more and more information is incorporated into the system through time, the estimates of the parameters consistently improve. The filter initialises with a

prescribed starting value, then adjusts it within the specified uncertainty to improve the fit to the measurement. There is a trade-off between the spatial and temporal smoothing hyper-parameters. To address this potential issue, we conducted a few preliminary runs to determine a reasonable spatial smoothing hyper-parameter that reliably produces both a smooth slip distribution and good fit to the data. We then employ this value with a small covariance thereby only allowing its value to be minimally changed by the Kalman filter, while using a large covariance for the temporal smoothing hyper-parameter which allows for a broad range of adjustments when necessary during the updating step. We finally obtain the initial guess for the temporal smoothing hyper-parameter, through a recursive algorithm where each previous estimate is updated to the current input, until model results from the current estimates and previous ones converge to within a specified tolerance (see ref. 23).”

Theoretically, the spatial weighting will have some influence on the model fits with data. However, for some stations that were installed after the 2004 rupture, the model fits with the data wouldn't change significantly because of the shift as explained before. In the meanwhile, the large uncertainty of the data would force the filter to not pay attention during the updating step, thus will not significantly improve the model fits.

Reference

Freed, A. M., G. Hirth, and M. D. Behn, Using short-term postseismic displacements to infer the ambient deformation conditions of the upper mantle, *J. Geophys. Res.*, 117, B01,409, doi:10.1029/2011JB008,562, 2012.

We cite the paper in the main content, line 178.

We thank Reviewer #2 for his or her thoughtful comments.

Reviewers' comments:

Reviewer #2 (Remarks to the Author):

The authors use an advanced new method to infer both time-dependent afterslip and effective viscosity of the Sumatran mantle wedge using a decade of GPS data and tide gauge data. This method was recently applied to the postseismic relaxation of the 2016 Kumamoto, Japan, earthquake. The authors build on the success of that study and demonstrate that in another well instrumented area — the Sumatran subduction zone — they are similarly able to infer rheological parameters. Their new method is powerful because it allows direct inference of 3D viscosity structure without the procedure of running numerous forward models typically used in postseismic relaxation studies. The identification of both the transient and long term (Maxwell) viscosities in this study is a remarkable achievement in itself, as all previous studies of post-earthquake relaxation that use this parameterization can well resolve only the transient viscosity. The results carry important implications for the physical state of the Sumatran mantle wedge and the micromechanical processes that underlie time-dependent rock creep. I recommend publication with minor revisions.

Specific comments

The finding that a Burgers body rheology, rather than a non-linear rheology with stress exponent of 2 or higher, best explains the time-dependent effective viscosity pattern is the most impressive conclusion of the study in my view. It should be described in the abstract if space permits.

Figure 5 is an impressive display of the relative importance of afterslip and viscoelastic relaxation on the postseismic deformation. The contribution of afterslip generally dwarfs that of viscoelastic relaxation. This raises the question of the tradeoff between the two processes, as a small change in the invoked afterslip would seemingly strongly alter the inferred volume strains and hence viscoelastic parameters. This appears to be thoroughly addressed with the resolution tests presented in Supplementary Figure 2, but any additional remarks about the need for afterslip in the main text (e.g., data is poorly fit without it) would make this tradeoff less of a concern.

In the previous Nature submission, eqn 1 was derived in the supplement (its 'Effective viscosity of a simple bi-viscous (Burgers) rheology' section). In the present version it appears out of thin air! Could the derivation be restored to the ms?

Supplementary Figure 5. There is systematic scatter in the tide gauge measurements, e.g., the dip in elevation from 2010 to 2011 seen at several stations. Any explanation for this signal, which is unrelated to the processes you're trying to model, would help the reader better understand non-tectonic signals in this data.

lines 439-440. time dependence features -> time dependent features

line 442. Pa S -> Pa s

Reviewer #3 (Remarks to the Author):

Qiang et al. present a comprehensive model analysis of postseismic deformation of multiple great megathrust earthquakes along the Sumatra-Andaman megathrust focused on establishing spatially variable and time-dependent viscosities and thus illuminate the underlying mantle rheology. This is a substantial and high-quality manuscript bringing a number of methodological advances to bear on this problem and thus complementing earlier model investigations. This rather unique contribution comes to some important and possibly controversial conclusions regarding the first

order rheology (linear biviscous, not power-law \pm early transient relaxation) and laterally heterogeneous strength of the mantle wedge.

As this paper has already gone through a first round of reviews and revisions, I am focusing on the degree to which the authors have addressed the review comments and a few related larger issues. I have read the reviews and author responses to the revised manuscript, and find that the authors have generally done a thorough job trying to address the reviewers' constructive concerns. I agree that given the data limitations, considering the transients from these four recent megathrust events together makes sense. While I list several detailed suggestions regarding some of the new material below, these do not keep me from recommending acceptance of the paper once these have been addressed.

With regards to the question of Rev1 about power-law rheology and related text in lines 210-225, I have a follow-up question. In a power-law material effective viscosities directly depend on stress and thus the coseismic stress increases, if background stress levels are low (e.g., see Freed et al., 2006 EPSL Fig. 7). This effect should be particularly strong for great megathrust earthquake cycles, given the large reach and magnitude of the stress changes. Do volumes of inferred reduced viscosity (Figure 6) correlate with those of increased shear-stress amplitudes (somewhat represented by the synthetic strain case in Figure S2a it seems). Or can this be ruled out lending further support to the suggestion of a dominant linear diffusion-creep rheology at steady state? One can't tell either way from visual inspection alone, but maybe you could make a scatter plot of cuboid stress vs. inferred viscosity for all (resolved) cuboids, possibly highlighting the data points that represent the initial coseismic stress and transient viscosity or using symbol color to indicate time (or maybe more similar to Fig. 8 plotting stress vs. strain rate). I realize this gets a little complicated, as we can also expect viscosity to go down with depth due to increasing temperature (e.g., Freed et al., 2016 EPSL).

Itemized Comments:

- The abstract makes no mention of the finding of laterally heterogeneous viscosity structure. I would try to find a way to fold this in.
- Line 27: Consider rewording the lead sentence, to get rid of the somewhat redundant "great and giant" and awkward "disturb and drag". Maybe something like "Great earthquakes generate large stress perturbations across a wide area in both the adjoining crust and upper mantle."
- Line 32: I would refrain from citing two recent papers by coauthors of this manuscript on such a broad statement that doesn't require a reference. Readers may consider this to be a case of somewhat gratuitous self-referencing? These fine papers are appropriately cited later on.
- Line 43: the current wording might suggest to some that more than four events are included in the model. However, I think it's just those four, with the contributions of the 2012 M8.6 and its postseismic deformation (and presumably offsets from other small earthquake) having been removed.
- Line 46: Add "far-field continuous and campaign GPS measurements" to this initial list of data constraints.
- Line 61: To address Rev1 concern, I would make clear here in the text that "the postseismic time series" refers to "(observed motions corrected for interseismic, seasonal, and earthquake related deformation, see Methods)".
- Line 62: For clarity, change "on the fault" to "on the subduction thrust" or "on the megathrust"?
- Line 65: Cite Hughes et al. (2010) on poroelastic deformation, whose models do indeed suggest short-term and mostly near-field contributions from this process.
- Line 72-73: Have you tested this statement about the ocean relaxation not being important? Forward modeling studies show that the deformation data are also impacted by the mantle rheology below the adjoining oceanic crust (e.g., Wiseman et al., 2015; Hu et al., 2016).
- Line 83: Penalizing dilatation makes sense, but remember the results by Ogawa and Heki (2007) and other groups modeling the GRACE data to partly reflect "relaxation of coseismic dilatation and compression by the diffusion of supercritical H₂O abundant in the upper mantle". Maybe briefly discuss this in the Discussion?
- Line 130-135: It is particularly challenging to resolve changing properties along the margin. What range of viscosities do you obtain when considering the obtained strains in those poorly

resolved volumes? I am curious to know if a 3D forward model that features no viscosity contrast along arc (using your favored bi-viscous rheology parameters), can produce equally satisfactory fits to the data. Maybe this could be included to accompany the model comparisons with the "dynamic" VISCO1D model around lines 130-135).

- Line 165, Equation 1, and related text in rebuttal and manuscript: The concept of a transient time scale is useful, but in the Burgers body the contribution of the Kelvin element is also determined by its elastic component and in studies of transient strain the transition is expected to occur at a certain finite strain, not time. Are Kelvin and Maxwell component rigidities provided? Okay, I see you do something like this in the last section of the Methods.

- Line 217, "soft slip system": Are you referring to dislocation glide or grain boundary sliding during the transient phase?

- Line 222: Add linear for "and linear steady-state creep". The interpretation of dominant linear diffusion creep during steady-state earthquake cycle deformation has quite important implications for various related issues, including the interpretation of seismic anisotropy and geodynamics.

- Line 400: Figure S2 is helpful. However, I am curious about the choice of the synthetic input strain in the "checkerboard test" being the spatial distribution of the coseismic stress changes, rather than a regular broad pattern of alternating strain. I realize that checkerboard tests are always somewhat limited and potentially biased by the choice of synthetic patterns, but this may be somewhat confusing to readers.

- Table 1: Not sure if it is useful to have all these cuboid viscosities in main body of the manuscript. Maybe provide a mean and range for your study together with those of previous studies, and list the detailed listing in supplement?

- Line 430: To be honest, I don't fully understand this new treatment and the need for "background stress and strain rate of the mantle wedge flow" for this estimate. As we only deal with the separated postseismic transients and are not considering a stress-dependent rheology why do we need these?

- Line 468: Fig. 8, not 7

- Line 686, Fig. 3: Is the continental lithosphere in these models really 100 km thick? That seems a bit high for a tectonically active backarc region. As it seems that inverted afterslip reaches deep down to the base of the modeled continental lithosphere, this may minimize the contribution from viscous relaxation to the model predictions as indicated by Fig. 4.

- Line 691, Fig. 4: Light blue vectors (not green) are horizontal from afterslip. Indicate "total" time period (2004 – 2014) in caption.

- Line 696, "Note that the scales for afterslip and viscoelastic flow displacement are different": That is not a good idea at all, making it impossible to compare observations and model predictions and to assess how the combination of afterslip and viscous strain can produce the observed deformation. However, is something wrong with this figure and/or the scale labels? According to this figure, viscous relaxation contributes minimally to the deformation, even in the far-field (Thailand/Malaysia), in great contrast to all previous studies (e.g., see Fig. 8-10 in Wiseman et al.). However, you write in the text "the far-field postseismic displacements are almost completely dominated by widespread viscoelastic flow in the mantle wedge (Fig. 4)".

- Line 700, Figure 5: Here also, the viscous contribution in the inversion and the VISCO1D predictions are very small compared to the observations for this event, suggesting minimal contributions.

Roland Bürgmann

Response to reviewers for “Transient rheology of the Sumatran mantle wedge revealed by a decade of great earthquakes” (Paper NCOMMS-17-03821A)
Qiang Qiu, James D. P. Moore, Sylvain Barbot, Lujia Feng, Emma M. Hill, under second revisions with Nature Communications

Reviewer #2 (Remarks to the Author):

The authors use an advanced new method to infer both time-dependent afterslip and effective viscosity of the Sumatran mantle wedge using a decade of GPS data and tide gauge data. This method was recently applied to the postseismic relaxation of the 2016 Kumamoto, Japan, earthquake. The authors build on the success of that study and demonstrate that in another well instrumented area — the Sumatran subduction zone — they are similarly able to infer rheological parameters. Their new method is powerful because it allows direct inference of 3D viscosity structure without the procedure of running numerous forward models typically used

in postseismic relaxation studies. The identification of both the transient and long term (Maxwell) viscosities in this study is a remarkable achievement in itself, as all previous studies of post-earthquake relaxation that use this parameterization can well resolve only the transient viscosity. The results carry important implications for the physical state of the Sumatran mantle wedge and the micromechanical processes that underlie time-dependent rock creep. I recommend publication with minor revisions.

We are very grateful for your supportive comments about our manuscript.

Specific comments

The finding that a Burgers body rheology, rather than a non-linear rheology with stress exponent of 2 or higher, best explains the time-dependent effective viscosity pattern is the most impressive conclusion of the study in my view. It should be described in the abstract if space permits.

Yes, indeed a good suggestion. We rephrased the abstract at line 18 as follows.

“We show that the evolution of stress and strain rate following these earthquakes is better matched by a bi-viscous than by a power-law rheology model, and we estimate laterally heterogeneous transient and background viscosities on the order of $\sim 10^{17}$ and $\sim 10^{19}$ Pa s respectively.”

Figure 5 is an impressive display of the relative importance of afterslip and viscoelastic relaxation on the postseismic deformation. The contribution of afterslip generally dwarfs that of viscoelastic relaxation. This raises the question of the tradeoff between the two processes, as a small change in the invoked afterslip would seemingly strongly alter the inferred volume strains and hence viscoelastic parameters. This appears to be thoroughly addressed with the resolution tests presented in Supplementary Figure 2, but any additional remarks about the need for afterslip in the main text (e.g., data is poorly fit without it) would make this tradeoff less of a concern.

The trade-off between the two mechanisms is a challenge to resolve, but the vertical displacement field is key to discriminating between these mechanisms e.g., vertical displacements of station PSKI in Fig. 5 cannot be fit by a single mechanism. We add text to this effect at lines 151-154.

“...Bengkulu earthquake (Fig. 5). We note that the vertical displacements provide a key discriminant between these two mechanisms e.g., station PSKI in Fig. 5, often exhibiting the opposite sense of motion in the vertical.”

In the previous Nature submission, eqn 1 was derived in the supplement (its `Effective viscosity

of a simple bi-viscous (Burgers) rheology' section). In the present version it appears out of thin air! Could the derivation be restored to the ms?

These equations are for data reduction only and their physical basis is not relevant. We chose them for their simple form and their good match to the data. The relevant equations for the perturbation of a Burgers material under constant loading are more complicated and are shown in the "Phase diagram" section of the supplementary materials.

Supplementary Figure 5. There is systematic scatter in the tide gauge measurements, e.g., the dip in elevation from 2010 to 2011 seen at several stations. Any explanation for this signal, which is unrelated to the processes you're trying to model, would help the reader better understand non-tectonic signals in this data.

We have added sentences at lines 400 to 404 to put comments on these scatter measurements.

"We also noted a systematic scatter in the vertical displacement time series for tide gauges around Singapore over time period ~2010 to 2011 (Supplementary Fig. 5). We are not clear what may have caused this offset, but we saw significant amplitude decline of the Aviso time series over this time period, which results this offset across all the tide gauges in Singapore region."

lines 439-440. time dependence features -> time dependent features

Done.

line 442. Pa S -> Pa s

Done.

Reviewer #3 (Remarks to the Author):

Qiang et al. present a comprehensive model analysis of postseismic deformation of multiple great megathrust earthquakes along the Sumatra-Andaman megathrust focused on establishing spatially variable and time-dependent viscosities and thus illuminate the underlying mantle rheology. This is a substantial and high-quality manuscript bringing a number of methodological advances to bear on this problem and thus complementing earlier model investigations. This rather unique contribution comes to some important and possibly controversial conclusions regarding the first order rheology (linear biviscous, not power-law \pm early transient relaxation) and laterally heterogeneous strength of the mantle wedge.

We are grateful for these positive and helpful review comments.

As this paper has already gone through a first round of reviews and revisions, I am focusing on the degree to which the authors have addressed the review comments and a few related larger issues. I have read the reviews and author responses the revised manuscript, and find that the authors have generally done a thorough job trying to address the reviewers' constructive concerns. I agree that given the data limitations, considering the transients from these four recent megathrust events together makes sense. While I list several detailed suggestions regarding some of the new material below, these do not keep me from recommending acceptance of the paper once these have been addressed.

With regards to the question of Rev1 about power-law rheology and related text in lines 210-225, I have a follow-up question. In a power-law material effective viscosities directly depend on stress and thus the coseismic stress increases, if background stress levels are low (e.g., see Freed et al., 2006 EPSL Fig. 7). This effect should be particularly strong for great megathrust earthquake cycles, given the large reach and magnitude of the stress changes. Do volumes of inferred reduced viscosity (Figure 6) correlate with those of increased shear-stress amplitudes (somewhat represented by the synthetic strain case in Figure S2a it seems). Or can this be ruled out lending further support to the suggestion of a dominant linear diffusion-creep rheology at steady state? One can't tell either way from visual inspection alone, but maybe you could make a scatter plot of cuboid stress vs. inferred viscosity for all (resolved) cuboids, possibly highlighting the data points that represent the initial coseismic stress and transient viscosity or using symbol color to indicate time (or maybe more similar to Fig. 8 plotting stress vs. strain rate). I realize this gets a little complicated, as we can also expect viscosity to go down with depth due to increasing temperature (e.g., Freed et al., 2016 EPSL).

We have included a new supplementary Fig. 8 (the scatter plot, and also map view of initial coseismic stress and transient viscosity of cuboids), with accompanying text at lines 296-299, to address this question.

“In addition, the estimated magnitude of the transient and steady-state viscosities of the resolved cuboids are not correlated with the magnitude of the spatial distribution of the coseismic stress changes (Supplementary Fig. 8), further ruling out the possibility of the nonlinear power-law rheology.”

Itemized Comments:

- The abstract makes no mention of the finding of laterally heterogeneous viscosity structure. I would try to find a way to fold this in.

Done, at line 20.

“..., and we estimate laterally heterogeneous transient and background viscosities on the ...”

- Line 27: Consider rewording the lead sentence, to get rid of the somewhat redundant “great and giant” and awkward “disturb and drag”. Maybe something like “Great earthquakes

generate large stress perturbations across a wide area in both the adjoining crust and upper mantle.”

Fixed.

- Line 32: I would refrain from citing two recent papers by coauthors of this manuscript on such a broad statement that doesn't require a reference. Readers may consider this to be a case of somewhat gratuitous self-referencing? These fine papers are appropriately cited later on.

Fixed.

- Line 43: the current wording might suggest to some that more than four events are included in the model. However, I think it's just those four, with the contributions of the 2012 M8.6 and its postseismic deformation (and presumably offsets from other small earthquake) having been removed.

Fixed at lines 51-53.

“They are four $M_w \geq 7.8$ events including the 2004 M_w 9.2 Sumatra–Andaman⁴, the 2005 M_w 8.6 Nias–Simeulue⁵, the 2007 M_w 8.4 Bengkulu⁶, and the 2010 M_w 7.8 Mentawai⁷ earthquakes.”

- Line 46: Add “far-field continuous and campaign GPS measurements” to this initial list of data constraints.

Fixed.

- Line 61: To address Rev1 concern, I would make clear here in the text that “the postseismic time series” refers to “(observed motions corrected for interseismic, seasonal, and earthquake related deformation, see Methods)”.

Fixed.

- Line 62: For clarity, change “on the fault” to “on the subduction thrust” or “on the megathrust”?

Fixed.

- Line 65: Cite Hughes et al. (2010) on poroelastic deformation, whose models do indeed suggest short-term and mostly near-field contributions from this process.

Included.

- Line 72-73: Have you tested this statement about the ocean relaxation not being important? Forward modeling studies show that the deformation data are also impacted by the mantle rheology below the adjoining oceanic crust (e.g., Wiseman et al., 2015; Hu et al., 2016).

We agree that the oceanic mantle relaxation may have some effect on our observations. However, our measurements are all limited to land-based GPS stations, therefore, we cannot resolve the viscous strain of cuboids in the oceanic mantle in our inversions. Introducing more free parameters into the inversion, especially in a region we cannot constrain, will decrease the stability of the solution. Considering the time frame of the observation in our study, and due to the lack of constraints, we took the decision not to include the oceanic mantle. We intend on addressing that question in future work when we will also examine gravitational effects.

- Line 83: Penalizing dilatation makes sense, but remember the results by Ogawa and Heki (2007) and other groups modeling the GRACE data to partly reflect “relaxation of coseismic dilatation and compression by the diffusion of supercritical H₂O abundant in the upper mantle”. Maybe briefly discuss this in the Discussion?

This is precisely why we penalize, because it restricts but does not prevent. We have cited reference *Ogawa and Heki (2007)* and also accompanying texts at lines 483-484.

“This results in a majority deviatoric solution, but does not prevent some degree of dilatation in the cuboid volumes⁵⁹.”

- Line 130-135: It is particularly challenging to resolve changing properties along the margin. What range of viscosities do you obtain when considering the obtained strains in those poorly resolved volumes? I am curious to know if a 3D forward model that features no viscosity contrast along arc (using your favored bi-viscous rheology parameters), can produce equally satisfactory fits to the data. Maybe this could be included to accompany the model comparisons with the “dynamic” VISCO1D model around lines 130-135).

We agree that resolving spatial variation of rheological properties along the margin is a challenging task, requiring smaller cuboids resulting in many more free parameters, which in turn requires a denser geodetic network to provide the constraints. With improved data coverage and facility, this is theoretically achievable. At the Sumatra subduction zone, unfortunately, the SuGAR network is not dense enough to resolve more details of spatial variation of the deep structure both in depth and along strike. We hope future high spatial and temporal resolution data sets e.g., InSAR, Gravity and a dense GPS network either here or elsewhere will allow us to uncover more details of the rheological structure.

For the poorly resolved cuboids, we simply cannot place any reasonable bounds on the effective viscosity.

We have included the model predictions by VISCO1D based on our preferred bi-viscous rheology parameters of cuboid 2 in Fig.5. Our inversion and the forward model are consistent with each other and the data, within the uncertainty of the model parameters.

- Line 165, Equation 1, and related text in rebuttal and manuscript: The concept of a transient time scale is useful, but in the Burgers body the contribution of the Kelvin element is also determined by its elastic component and in studies of transient strain the transition is expected to occur at a certain finite strain, not time. Are Kelvin and Maxwell component rigidities provided? Okay, I see you do something like this in the last section of the Methods.

Yes, we've done this in the Methods section. We also made it clear at lines 240-241.

“...for the background viscosity η_M the transient viscosity η_K , and the time scale of the transient $\tau_K = \frac{\eta_K}{G_K}$, where G_K is the shear modular of the Kelvin body.”

- Line 217, “soft slip system”: Are you referring to dislocation glide or grain boundary sliding during the transient phase?

We are referring the “grain boundary sliding during the transient phase”. We have made it clear at lines 291-292.

“...motion of the soft slip system (grain boundary sliding during the transient phase) of ...”

- Line 222: Add linear for “and linear steady-state creep”. The interpretation of dominant linear diffusion creep during steady-state earthquake cycle deformation has quite important implications for various related issues, including the interpretation of seismic anisotropy and geodynamics.

Fixed.

- Line 400: Figure S2 is helpful. However, I am curious about the choice of the synthetic input strain in the “checkerboard test” being the spatial distribution of the coseismic stress changes, rather than a regular broad pattern of alternating strain. I realize that checkerboard tests are always somewhat limited and potentially biased by the choice of synthetic patterns, but this may be somewhat confusing to readers.

It is very difficult to pick the ideal synthetic test, with every researcher preferring their own method. The spirit of the checkerboard test is basically to guide us as to where and how much we can resolve the synthetic input, depending on the design patterns of the checkerboards e.g., the spatial arrangements, checkerboard sizes. These kinds of tests can be useful but also have limitations and may introduce biases, depending on the choice of the synthetic patterns. The basic result is that where we have more data, the resolution is better. In our case, we have decided to use the spatial pattern of the coseismic stress changes to design the synthetic input for the checkerboard tests, instead of using a broad pattern of alternating strain because this more closely matches the physical assumptions employed in the dynamic forward models. Where we have the large stress concentrations, we expect viscous strain to occur. Thus, using this spatial pattern as the synthetic inputs can help us to understand our final inversion results and build confidence in the best resolved regions. We therefore thought it was the most

meaningful test, not only for us, but also for reader to best understand the results.

- Table 1: Not sure if it is useful to have all these cuboid viscosities in main body of the manuscript. Maybe provide a mean and range for your study together with those of previous studies, and list the detailed listing in supplement?

We present our estimated effective viscosity ranges from our inversion (lines 20 and 249), and highlight the consistence with the published viscosity range in the main content. We include the table to list the estimated rheological parameters and the published ones to allow readers to directly compare them, and also to look at the possible spatial variations of the viscosity as well. Therefore, we leave it up to the Editors discretion as to whether it is best located in the main content or the Supplementary material.

- Line 430: To be honest, I don't fully understand this new treatment and the need for "background stress and strain rate of the mantle wedge flow" for this estimate. As we only deal with the separated postseismic transients and are not considering a stress-dependent rheology why do we need these?

Before the earthquake occurred, we assume that the mantle wedge flow was at a steady-state stage. Thus, we assume that the background stress and the strain rate were at an equilibrium condition. When the megathrust ruptured, it disturbed and accelerated the mantle flow for years and decades to come. When the extra coseismic stress is completely released to be zero, then it will go back to the steady-state equilibrium condition again. Thus, to get the steady-state viscosity we must necessarily include the background stress and strain rates in our calculation.

- Line 468: Fig. 8, not 7

Fixed.

- Line 686, Fig. 3: Is the continental lithosphere in these models really 100 km thick? That seems a bit high for a tectonically active backarc region. As it seems that inverted afterslip reaches deep down to the base of the modeled continental lithosphere, this may minimize the contribution from viscous relaxation to the model predictions as indicated by Fig. 4.

Determine the actually boundary between the brittle and the ductile region is a challenge. As an example, we can see different elastic thicknesses have been proposed to model the postseismic deformation following the Mw 9.2 Sumatra-Andaman earthquake (*Broerse et al. (2015), Hu and Wang 2012, Pollitz et al. (2006, 2008), Panet et al. (2010), Lubis et al. (2012), Han et al. (2008), Hoechner et al. (2011), Gunawan et al. (2014)*). In our case, we allow the depth of the cuboids to varying along strike instead of arbitrarily selecting one depth cut off for the whole area. Besides, the lithosphere/asthenosphere boundary (LAB) study through shear-wave receiver function technique (*Kumar et al. (2007), Figure 1*) indicates that the LAB thickness at the Sumatra subduction zone is $\sim < 120$ km. Thus, our final depth of the cuboids vary from ~ 50 km to 230 km, which falls in the range from previous publications (*Broerse et al. (2015), Hu and Wang 2012, Pollitz et al. (2006, 2008), Panet et al. (2010), Lubis et al. (2012)*),

Han et al. (2008), Hoechner et al. (2011), Gunawan et al. (2014).

- Line 691, Fig. 4: Light blue vectors (not green) are horizontal from afterslip. Indicate “total” time period (2004 – 2014) in caption.

Fixed.

- Line 696, “Note that the scales for afterslip and viscoelastic flow displacement are different”: That is not a good idea at all, making it impossible to compare observations and model predictions and to assess how the combination of afterslip and viscous strain can produce the observed deformation. However, is something wrong with this figure and/or the scale labels? According to this figure, viscous relaxation contributes minimally to the deformation, even in the far-field (Thailand/Malaysia), in great contrast to all previous studies (e.g., see Fig. 8-10 in Wiseman et al.). However, you write in the text “the far-field postseismic displacements are almost completely dominated by widespread viscoelastic flow in the mantle wedge (Fig. 4)”.

We have fixed the scale issue in Fig. 4.

Yes, we notice that the afterslip prediction for the horizontal components is larger than the viscoelastic flow prediction in the far-field Thailand/Malaysia region. This is because our geodetic data network, and computational difficulties did not allow us to pave cuboids all the way to underneath these places where far-field GPS stations are. Instead, we have to limit the cuboids around the fault zones. However, as the shallower afterslip cannot explain the far-field vertical components, the deep viscoelastic flow takes it over and dominates the contribution for the vertical measurements e.g., tide gauge vertical measurements of the west coast of Malaysia and Singapore. We hope future dense geodetic observations like InSAR and gravity can solve this issue in our following studies. We modify the text at lines 135-136.

“..., while the far-field postseismic displacements (e.g., the tide gauges along the coast of Malaysia and Singapore) are almost completely...”

- Line 700, Figure 5: Here also, the viscous contribution in the inversion and the VISCO1D predictions are very small compared to the observations for this event, suggesting minimal contributions.

The sample stations shown in Fig. 5 (Fig. 2 and Fig. 4 for geographic locations) are located above the downdip part of the 2007 Bengkulu earthquake. Therefore, they should experience more fast motion from the deep afterslip on the megathrust, as indicated by the inversion results and VISCO1D prediction shown in Fig. 5.

Roland Bürgmann

Reviewers' comments:

Reviewer #3 (Remarks to the Author):

The authors have done a good job addressing most all of my suggestions and questions. I recommend accepting the paper for publication in NCOMMS, but again have a concern about the (otherwise nicely revised) Figure 4.

For some reason, the model predictions from the viscoelastic relaxation are now effectively missing or invisible, which is inconsistent with the description in the text ("the far-field postseismic displacements ... are almost completely dominated by widespread viscoelastic flow in the mantle wedge (Fig. 4).") and also inconsistent with vectors shown in Figure 3.

Comparing this with the old Figure 4, I get the impression that the VE prediction vectors are effectively non-existent everywhere except in central Sumatra (they seem to disappear because of the corrected scale), but the afterslip prediction now looks different too (afterslip vectors in N Sumatra and Thailand are now substantially smaller). I hope this is a simple plotting mistake, rather than a serious issue with the models.

I hope that can be clarified/corrected and am looking forward to seeing this paper in published soon.

Response to Reviewers' comments:

Reviewer #3 (Remarks to the Author):

The authors have done a good job addressing most all of my suggestions and questions. I recommend accepting the paper for publication in NCOMMS, but again have a concern about the (otherwise nicely revised) Figure 4.

We are grateful for these positive and helpful review comments.

For some reason, the model predictions from the viscoelastic relaxation are now effectively missing or invisible, which is inconsistent with the description in the text ("the far-field postseismic displacements ... are almost completely dominated by widespread viscoelastic flow in the mantle wedge (Fig. 4).") and also inconsistent with vectors shown in Figure 3.

Comparing this with the old Figure 4, I get the impression that the VE prediction vectors are effectively non-existent everywhere except in central Sumatra (they seem to disappear because of the corrected scale), but the afterslip prediction now looks different too (afterslip vectors in N Sumatra and Thailand are now substantially smaller). I hope this is a simple plotting mistake, rather than a serious issue with the models.

I hope that can be clarified/corrected and am looking forward to seeing this paper in published soon.

Plotting the viscoelastic and afterslip contributions alongside the GPS time series from the inversion in map view is a difficult proposal. This is because each mechanism has its own timescales associated with it. The viscoelastic deformation has a short-lived transient phase (a year or so), and a steady-state phase with a viscosity (and corresponding timescales) one magnitude higher, resulting in far slower strain rate after the initial transient phase. This in turn means the intermediate-time (several years) contribution to the accumulated surface displacement at the later time stage can be less than the afterslip, but will continue for longer e.g., solid blue curves in Figure 5. Therefore, as time goes on, the accumulated surface displacement difference between the GPS time series and the viscoelastic flow contribution becomes larger (e.g., Figure 5, data and solid blue curves). If we plot the final cumulative displacements of the observations, afterslip and viscoelastic flow contributions on the same scale, it can be hard to distinguish the individual components clearly. This is why we initially plotted the displacements 6 months after the earthquakes, as shown in Figure 4 in the last submission. This also explains why the vectors look different when compared to Figure 3 – they are also cumulative displacements, but over a different time scale, as explained in the respective captions. In addition, stations in the northern Sumatra and Thailand have offsets due to their deployment after the 2004 Sumatra-Andaman earthquake, missing the early fast motions; thus the early afterslip model prediction is even smaller when compared to the incomplete GPS data. In summary, plotting cumulative displacements at a given time interval tends to bias the figure towards one particular mechanism as they operate over different timescales.

To maximise our ability to visualise the individual components of afterslip and viscoelastic flow, we turn to the vertical component of deformation. We show that the viscoelastic flow not only provides a significant contribution to the vertical displacements in the near-field, but also dominates the vertical displacement of the far-field GPS and Tide gauges. The resolution matrix of strain components e_{11} , e_{13} and e_{33} (Supplementary Figure S8) show an increasing resolution towards the far-field GPS and Tide gauges, suggesting that the far-field tide gauges play an important role in constraining the cuboids, and a well resolved vertical component when we explore the far-field. Therefore, we have updated Figure 4 to illustrate the vertical component of afterslip and viscoelastic flow at the GPS and tide gauges, as well as sample stations where we obtain significant viscoelastic flow contributions, with the full time series for each site given in supplementary Figure 4 & 5. The former Figure 4 has now been moved to the supplementary material for interested readers. We have modified the text at line 118-123, as shown below:

“..., while the widespread viscoelastic flow in the mantle wedge contributes significantly to the vertical displacements not only for the near-field, but also dominates the far-field vertical postseismic displacements (e.g., the tide gauges along the coast of Malaysia and Singapore; Fig. 4, we also show the horizontal components of the early time stage in Supplementary Fig. 6).”

We have also updated the caption of Figure 4 at lines 737-748.

“Our estimated cumulative displacements for the vertical component of the mechanical decomposition of afterslip and viscoelastic flow at all geodetic stations over the whole time period (from 2005 to 2014). White vectors represent the cumulative vertical displacements at GPS stations, and tide gauges. Light blue vectors show the cumulative vertical displacements due to afterslip on the megathrust. Red vectors indicate the cumulative vertical displacements from viscoelastic flow in the mantle wedge. Sample stations (with names are labelled in the map) where the viscoelastic flow contributes significantly to the displacement are shown in more detailed time series. Red, blue and black curves represent the viscoelastic flow, afterslip prediction and data, respectively. Due to the large variation of the tide gauge time series, we plot the cumulative displacements from least-squares fits, for the full time series of GPS and original tide gauges are shown in the supplementary Figure 4 & 5.”

REVIEWERS' COMMENTS:

Reviewer #3 (Remarks to the Author):

Following the last round of revisions, I would like to recommend publication of this manuscript.

Roland Burgmann